# On the Sample Complexity of Differentially Private Policy Optimization

**Yi He**
Wayne State University
yihe@wayne.edu

**Xingyu Zhou**
Wayne State University
xingyu.zhou@wayne.edu

## Abstract

Policy optimization (PO) is a cornerstone of modern reinforcement learning (RL), with diverse applications spanning robotics, healthcare, and large language model training. The increasing deployment of PO in sensitive domains, however, raises significant privacy concerns. In this paper, we initiate a theoretical study of differentially private policy optimization, focusing explicitly on its sample complexity. We first formalize an appropriate definition of differential privacy (DP) tailored to PO, addressing the inherent challenges arising from on-policy learning dynamics and the subtlety involved in defining the unit of privacy. We then systematically analyze the sample complexity of widely-used PO algorithms, including policy gradient (PG), natural policy gradient (NPG) and more, under DP constraints and various settings, via a unified framework. Our theoretical results demonstrate that privacy costs can often manifest as lower-order terms in the sample complexity, while also highlighting subtle yet important observations in private PO settings. These offer valuable practical insights for privacy-preserving PO algorithms.

## 1 Introduction

Policy Optimization (PO) such as REINFORCE [1, 2], proximal policy optimization (PPO) [3] and group relative policy optimization (GRPO) [4] has gained increasing interest recently across various applications. Due to its popularity, there is a rich literature that provides various theoretical understandings of different PO methods (e.g., iteration or sample complexity [5–8]).

As PO becomes increasingly prevalent in real-world applications, privacy concerns are emerging as a critical challenge. For instance, in personalized medical care, patient interactions—where the state represents medical history, the action corresponds to prescribed medication, and the reward reflects treatment effectiveness—constitute sensitive data that must be protected. Similarly, in RL-based training of large language models (LLMs), user prompts may contain private information that requires protection. In fact, recent empirical work has shown that standard GRPO indeed has privacy leakage issues [9]. Hence, addressing these privacy concerns is essential for ensuring the responsible deployment of PO methods in sensitive domains.

**Contribution.** In this paper, we initiate the theoretical study of differentially private policy optimization, focusing on the central question: *What's the sample complexity cost induced by differential privacy in PO?* To this end, we first carefully define a suitable notion of differential privacy (DP) [10] for PO, highlighting its distinctions from the standard DP definitions used in supervised learning. These differences stem from the unique learning dynamics and the notion of the privacy unit in PO. Then, we propose a meta algorithm for private PO, which enables us to study private policy gradient with REINFORCE (DP-PG), private natural policy gradient (NPG) [11] (DP-NPG), and private version of REBEL recently proposed in [12] (DP-REBEL) in a unified perspective. Moreover, for DP-NPG and DP-REBEL, we further reduce PO to a sequence of private regression problems, thus allowing us to

39th Conference on Neural Information Processing Systems (NeurIPS 2025).

leverage various well-established results in private estimation and supervised learning. Throughout this process, we not only highlight the difference between `DP-PG` and `DP-NPG`, but also uncover some subtleties when applying private regression results within the current analytical framework of PO. The key takeaway from our theoretical results is that the privacy cost can often appear as lower-order terms in the overall sample complexity. Meanwhile, it is worth noting that structural properties of the underlying problem can further improve both statistical and computational efficiency.

**Related work.** We mainly discuss the most relevant work here and relegate a detailed discussion to Appendix A. The very recent work [13] studies private PG, but mainly from an empirical perspective without sample complexity bounds. From the online regret perspective, optimistic PPO has been studied in the tabular case [14] and linear case [15], respectively. In contrast, we aim to consider general function classes from the optimization perspective. We also note that NPG with a softmax or log-linear policy is equivalent to PPO. The authors of [16] study private policy evaluation, which aims to evaluate a given policy rather than finding the best policy in policy optimization. From an application perspective, [17] applies private PPO for LLM alignment via reinforcement learning from human feedback (RLHF).

## 2 Preliminaries

**Policy optimization (PO).** In this work, instead of considering a general Markov decision process (MDP), we focus on the simpler bandit formulation, which allows us to easily demonstrate the key ideas. We note that generalizing it to MDP is standard, as done in the literature [6, 18]. This bandit formulation already captures many interesting real-world applications, such as personalized medical care [19] and alignment/reasoning training in large language models (LLMs) [20]. In particular, given an initial state $x \in \mathcal{X}$ (e.g., a medical status or a prompt in LLMs) sampled from a distribution $\rho$, an action $y \in \mathcal{Y}$ (e.g., a medical prescription or a response in LLMs) is generated according to a policy $\pi$ and a reward $r(x, y) \in [-R_{\mathsf{max}}, R_{\mathsf{max}}]$ is observed. In policy optimization, we parameterize the policy $\pi$ by $\pi_\theta$ with $\theta \in \Theta = \mathbb{R}^d$ (e.g., a neural network), and the goal is to leverage interactions (e.g., sample trajectories) to find an optimal policy that maximizes the following objective:

$$J(\pi_\theta) = J(\theta) := \mathbb{E}_{x\sim\rho, y\sim\pi_\theta(\cdot|x)}\left[r(x, y)\right].$$

**Vanilla policy gradient (PG).** One simple and direct approach to solving the above policy optimization problem is via vanilla policy gradient, i.e., $\theta_{t+1} = \theta_t + \eta \nabla J(\theta_t)$, where $\eta > 0$ is some learning rate, $\nabla J(\theta_t)$ is the gradient at step $t$, and $\theta_1$ is some initial value. The gradient can be written as follows by the classic policy gradient theorem

$$\nabla J(\theta) = \mathbb{E}_{x\sim\rho, y\sim\pi_\theta(\cdot|x)}\left[A^{\pi_\theta}(x, y)\nabla_\theta \log \pi_\theta(y|x)\right], \tag{1}$$

where $A^{\pi_\theta}(x, y) := r(x, y) - \mathbb{E}_{y'\sim\pi_\theta(y'|x)}r(x, y')$ is the advantage function.

**Natural policy gradient (NPG).** Another approach to solving the PO problem is natural policy gradient (NPG) [11], which uses the Fisher information matrix as the preconditioner to account for the geometry. Specifically, the NPG update is given by $\theta_{t+1} = \theta_t + \eta F_\rho^\dagger(\theta_t)\nabla J(\theta_t)$, where $F_\rho(\theta) := \mathbb{E}_{x\sim\rho, y\sim\pi_\theta(\cdot|x)}[\nabla_\theta \log \pi_\theta(y|x) \log \pi_\theta(y|x)^\top]$ is the expected Fisher information matrix (with superscript $\dagger$ being the Moore-Penrose pseudoinver) and $\nabla J(\theta_t)$ is the same as before. An equivalent way to write the above update is the following [5]

$$\theta_{t+1} = \theta_t + \eta \cdot w_t, w_t \in \operatorname*{argmin}_w \mathbb{E}_{x\sim\rho, y\sim\pi_{\theta_t}(\cdot|x)}\left[\left(A^{\pi_{\theta_t}}(x, y) - w^\top \nabla \log \pi_{\theta_t}(y|x)\right)^2\right], \tag{2}$$

which essentially reduces PO to a sequence of regression problems.

**Regression to Relative Reward Based RL (REBEL).** Recently, a simple and scalable PO algorithm called `REBEL` is proposed in [12], which also reduces PO to a sequence of regression problems, but now over *relative reward difference*, motivated from the DPO-style reparameterization trick in [21]. In the expected form, the update under `REBEL` is given by

$$\theta_{t+1} = \operatorname*{argmin}_\theta \mathbb{E}\left[\frac{1}{\eta}\left(\ln \frac{\pi_\theta(y\mid x)}{\pi_{\theta_t}(y\mid x)} - \ln \frac{\pi_\theta(y'\mid x)}{\pi_{\theta_t}(y'\mid x)}\right) - (r(x, y) - r(x, y'))\right]^2, \tag{3}$$

where the expectation here is over $x \sim \rho, y \sim \mu(\cdot|x), y' \sim \pi_{\theta_t}(\cdot|x)$, and $\mu$ can be either on-policy distribution $\pi_{\theta_t}$ or any offline reference policy.

**Sample complexity.** All the aforementioned ideal policy updates (e.g., full gradient) involve expectation, which is often difficult to compute in practice due to both statistical (e.g., without knowing $\rho$) and computational (e.g., averaging over all possible trajectories) issues. Thus, one needs to replace the expectation with a sample-based estimate by sampling a dataset of trajectories at each iteration from an underlying distribution. The sample complexity typically refers to the total number of sampled trajectories for finding an $\alpha$-optimal policy (i.e., $J(\pi^*) - J(\widehat{\pi}) \leq \alpha$).

In this paper, our ultimate goal is to formally introduce differential privacy (DP) into the problem of policy optimization and derive the sample complexity bounds under privacy constraints. To this end, we need to carefully define both privacy and samples in the private case, as discussed next.

## 3    Differential Privacy in Policy Optimization

In this section, we formally introduce DP to PO, highlighting some subtleties compared with standard DP in supervised learning problems. We first recall the standard DP definition with a *fixed* dataset.

**Definition 1** (Dwork et al. [10]). A randomized mechanism $\mathcal{M}$ satisfies $(\varepsilon, \delta)$-DP if for any adjacent datasets $D, D'$ differing by one record, and $\forall S \subseteq \text{Range}(\mathcal{M})$:

$$\mathbb{P}[\mathcal{M}(D) \in S] \leqslant e^{\varepsilon} \cdot \mathbb{P}[\mathcal{M}(D') \in S] + \delta.$$

This standard DP notion can be directly used in supervised learning problems with $D$ being a set of i.i.d samples $\{(x_i, y_i)\}_{i=1}^N$ from an unknown distribution and $\mathcal{M}(D)$ being the final model. This has been utilized in private empirical risk minimization (ERM) [22, 23] as well as private stochastic optimization (both convex and non-convex), e.g., Bassily et al. [24]. For example, the optimal excess population risk for stochastic convex optimization is $O_\delta(1/\sqrt{N} + \sqrt{d}/(N\varepsilon))$ for $(\varepsilon, \delta)$-DP, where $d$ is the dimension of the parameter space.

One may attempt to adopt the above notion directly to PO with the dataset $D$ being $\{(x_i, y_i)\}_{i=1}^N$ and $\mathcal{M}(D)$ being the final policy. However, this does not make too much sense because (i) there is no such a *fixed* dataset in PO as the actions are often sampled in the on-policy fashion, i.e., using the most recent policy; (ii) the neighboring relation of differing in one sample $(x_i, y_i)$ (i.e., privacy unit) actually does not hold as changing one sample will lead to difference in all future samples due to different policies onward. Thus, we need a new definition that can address the above two issues. Before proceeding, we consider two motivating examples to illustrate the subtlety.

**Example 1** (SFT vs. RL fine-tuning in LLMs). Consider a reasoning task in LLMs. With supervised fine-tuning (SFT), we are given a fixed dataset of pairs $\{(x_i, y_i)\}_{i=1}^N$ where $x_i$ is the prompt/question and $y_i$ is the correct answer. Standard DP is natural here, which ensures that changing one sample $(x_i, y_i)$ will not change the final policy too much. On the other hand, if one uses RL (e.g., PPO) to do the fine-tuning, then the given dataset consists of only prompts, as the answers are generated on the fly. So, a proper privacy unit here is to protect each prompt in the sense that changing one prompt will not change the policy too much.

**Example 2** (Supervised learning vs. RL for healthcare). In this case, to train a healthcare system, one can use a supervised learning approach by collecting a dataset of $\{(x_i, y_i)\}_{i=1}^N$ where $x_i$ is the medical status and $y_i$ is the recommended medicine. One can also adopt an RL approach (even in an online manner) where the dataset consists of a (stream) set of users/patients, each with a medical status $x_i$ sampled from a distribution $\rho$, while the recommendation $y_i$ can only be determined on the fly. The privacy protection is that changing one user/patient will not change the final policy too much.

To handle both scenarios, we borrow the idea from private online bandit and RL literature [25, 14], which essentially considers a set of "users" as the dataset. For instance, the dataset could be $N$ unique patients interacting with the learning agent, and each user has an initial state (e.g., medical status), which is distributed according to $\rho$. We can fix the "users" in advance (or arrive online) and the privacy unit is now for each patient, hence resolving both issues above. Meanwhile, the set of "users" can also represent $N$ (static) prompts in the fine-tuning of LLMs, with each "user" contributing one prompt. Note that although we use "users" to align with personalization application, this is still an item-level DP, as each "user" appears only once (as a patient or prompt). The learning agent can interact with each "user" to observe $(x, y)$ and $r(x, y)$ *dynamically*, i.e., on-the-fly. With the above notion of dataset, the privacy protection in PO is that changing one "user" in the dataset will not change the final policy too much, leading to the following definition.

**Definition 2** (DP in PO). Consider any policy optimization algorithm $\mathcal{M}$ interacting with a set $D$ of $N$ "users" and $\mathcal{M}(D)$ being the final output policy. We say $\mathcal{M}$ is $(\varepsilon, \delta)$-DP if for any adjacent datasets $D, D'$ differing by one "user", and $\forall S \subseteq \text{Range}(\mathcal{M})$:

$$\mathbb{P}[\mathcal{M}(D) \in S] \leqslant e^{\varepsilon} \cdot \mathbb{P}[\mathcal{M}(D') \in S] + \delta.$$

*Remark* 1. We emphasize that the above DP notion is defined for the problem PO rather than for a specific algorithm, analogous to the standard DP in statistical learning (e.g., supervised learning). In this paper, we aim to design some private variants of PO methods (`DP-PG`, `DP-NPG`, `DP-REBEL`) and analyze their sample complexity under the above privacy constraint.

## 4  A Meta Algorithm for Private PO

In this section, we present a meta algorithm for private PO, which builds upon a unified view of PG, NPG, and REBEL. We believe that this meta viewpoint is also interesting in the non-private case.

Our meta algorithm is given by Algorithm 1, which is essentially a batched one-pass algorithm. In particular, at each iteration $t$, the learner collects $m$ fresh samples by sampling from a distribution over $x$ and $y$. To be more specific, one can view each sample $(x_i, y_i, y_i')$ as generated via interaction with a new fresh "user", which provides the context/prompt $x_i$. Then, leveraging the dataset $D_t$ and a specific `PrivUpdate` oracle, the learner finds the next policy iteratively.

---

**Algorithm 1** A Meta Algorithm

---

1: **Input:** reward $r$, learning rate $\eta$, batch size $m$, and policy class $\pi_\theta$, `PrivUpdate` oracle, base policy $\mu$
2: Initialize $\theta_1 = 0$
3: **for** $t = 1, \ldots, T$ **do**
4:    Collect a *fresh* dataset $\bar{D}_t = \{(x_i, y_i, y_i')\}_{i=1}^m$ of size $m$ using the $\pi_{\theta_t}$ and $\mu$:

$$x_i \sim \rho, y_i \sim \mu(\cdot|x_i), y_i' \sim \pi_{\theta_t}(\cdot|x_i)$$

5:    For all $i \in [m]$, let $\widehat{A}_t(x_i, y_i) := r(x_i, y_i) - r(x_i, y_i')$ be the estimate of $A^{\pi_{\theta_t}}(x_i, y_i)$
6:    Call a `PrivUpdate` oracle on $D_t := \{(x_i, y_i, y_i', \widehat{A}_t(x_i, y_i))\}_{i=1}^m$ to find next policy $\theta_{t+1}$
7: **end for**

---

Under this one-pass algorithm design, we naturally have the following privacy guarantee, connecting standard DP (Definition 1) with DP in PO (Definition 2).

**Proposition 1.** *Suppose* `PrivUpdate` *satisfies $(\varepsilon, \delta)$-DP under Definition 1, then Algorithm 1 satisfies $(\varepsilon, \delta)$-DP in terms of Definition 2.*

This simply follows from our one-pass algorithm and (adaptive) parallel composition of DP, by noting that changing one "user" would only change one record in $D_t$ of a single $t \in [T]$.

*Remark* 2. Our meta algorithm can also be used in the online setting where a stream of $N$ "users" arrive sequentially. By the so-called *billboard lemma* [26], our meta algorithm also satisfies the commonly used *joint differential privacy* (JDP) in the literature of private online RL/bandits [25, 27, 14, 15, 28]. Roughly speaking, JDP guarantees that changing one "user" (say $u$) will not change all the actions prescribed to all other "users" except $u$, as well as the final policy.

For sample complexity, due to the batched one-pass algorithm over $N$ unique "users", the total number of sampled trajectories is simply $N = m \cdot T$, where each trajectory $(x_i, y_i, y_i')$ is from a fresh user. To put it in another way, for a fixed $N$, the key here is to balance between batch size $m$ and number of iterations $T$ so as to balance between the per-iteration accuracy and the total number of updates. This balance, in turn, depends on the specific choice of `PrivUpdate` oracle, which will be instantiated in the next sections for `DP-PG`, `DP-NPG`, `DP-REBEL`, respectively.

## 5  Differentially Private Policy Gradient

In this section, we propose `DP-PG` by building upon our meta algorithm and analyze its sample complexity bounds under different settings.

Our proposed `DP-PG` is Algorithm 1 with the instantiation of `PrivUpdate` as in Algorithm 2 below. In particular, it first computes an unbiased REINFORCE-style estimate (i.e., $\widehat{\nabla}_m J(\theta)$) of the full gradient $\nabla J(\theta_t)$ as in (1), using the $m$ trajectories in $D_t$. Then, a Gaussian noise is added with $\sigma^2$ depending on the privacy parameters of $\varepsilon$ and $\delta$. Finally, $\theta_{t+1}$ is obtained by updating the current policy $\theta_t$ along the direction of $\widetilde{g}_t$, scaled by a properly chosen learning rate $\eta$.

---

**Algorithm 2** `PrivUpdate` Instantiation for `DP-PG`

---

1: **Input:** dataset $D_t = \{(x_i, y_i, \widehat{A}_t(x_i, y_i))\}_{i=1}^m$, policy $\theta_t$, learning rate $\eta$, noise scale $\sigma$
2: **Output:** $\theta_{t+1}$
3: Compute gradient:

$$\widehat{\nabla}_m J(\theta) := \frac{1}{m} \sum_{i=1}^m \nabla_\theta \log \pi_{\theta_t}(y_i \mid x_i) \cdot \widehat{A}_t(x_i, y_i)$$

4: Add noise: $\widetilde{g}_t := \widehat{\nabla}_m J(\theta) + \mathcal{N}(0, \sigma^2 I)$
5: Output policy: $\theta_{t+1} = \theta_t + \eta \cdot \widetilde{g}_t$

---

By the standard Gaussian mechanism [29] and Proposition 1, we have the following privacy guarantee.

**Theorem 1** (Privacy guarantee). *Assume for any $x \in \mathcal{X}$ and $\theta \in \Theta$, there exists a constant $G$ such that $\|\nabla_\theta \log \pi_\theta(y \mid x)\| \leqslant G$. Then, setting $\sigma^2 = \frac{16 \log(1.25/\delta) R_{\max}^2 G^2}{m^2 \varepsilon^2}$ in Algorithm 2 ensures that* `DP-PG` *satisfies $(\varepsilon, \delta)$-DP as in Definition 2.*

The boundedness assumption of $G$ is satisfied by softmax policy as well as Gaussian policy [6]. In fact, they satisfy an even stronger condition in Assumption 1, to be discussed shortly.

Next, we aim to establish the sample complexity results of our `DP-PG` with Algorithm 2 for both first-order stationary point (FOSP) and global optimum convergence, respectively.

### 5.1 First-order Stationary Point Convergence

We start with the sample complexity bound for FOSP convergence. This result is not only of its own importance, but will also be useful for our later results on the global optimum convergence. We will consider the following general class of policies, which is widely studied in previous non-private work and also includes commonly used policies such as softmax and Gaussian policy [6].

**Assumption 1** (Lipschitz Smoothness (LS)). There exist constants $G, F > 0$ such that for every state $x \in \mathcal{X}$, the gradient and Hessian of $\log \pi_\theta(\cdot \mid x)$ of any $\theta \in \Theta$ satisfy

$$\|\nabla_\theta \log \pi_\theta(y|x)\| \leqslant G \text{ and } \|\nabla_\theta^2 \log \pi_\theta(y|x)\| \leqslant F.$$

*Remark* 3. For simplicity, as in previous work, we will often view $G$ and $F$ as constants $\Theta(1)$, hence omitted in the sample complexity bound. Moreover, we omit $\log(1/\delta)$ term by writing $O_\delta(\cdot)$.

**Theorem 2** (FOSP convergence). *Under the same setting of Theorem 1 and Assumption 1, there exists a proper parameter choices of $m$ and $\eta$, such that* `DP-PG` *achieves*

$$\mathbb{E}\left[\|\nabla J(\theta_U)\|^2\right] \leqslant O_\delta\left(\frac{1}{\sqrt{N}} + \left(\frac{\sqrt{d}}{N\varepsilon}\right)^{2/3}\right), \tag{4}$$

*where $\theta_U$ is uniformly sampled from $\{\theta_1, \ldots, \theta_T\}$.*

*Remark* 4. We can see that the first term in (4) matches the previous non-private term, i.e., for an accuracy of $\alpha$, the sample complexity is $O(1/\alpha^4)$ [6]; Second, the privacy cost is a lower order additive term (for constant $\varepsilon$ and $d$), i.e., the additional sample complexity due to privacy is $O_\delta\left(\frac{\sqrt{d}}{\alpha^3 \varepsilon}\right)$.

### 5.2 Global Optimum Convergence

We now turn our focus to the global optimum convergence in the sense of average regret, i.e., $J^* - \frac{1}{T} \sum_{t=1}^T \mathbb{E}\left[J(\theta_t)\right]$. Following the non-private work [6], we will also consider two different scenarios and aim to establish the corresponding sample complexities in the private case.

In the first scenario, in addition to Assumption 1, we further assume the following two conditions on the policy class, both of which are commonly used in the non-private case. The first condition is the so-called *Fisher-non-degenerate policy*, formally defined below.

**Assumption 2** (Fisher-non-degenerate, adapted from Assumption 2.1 of Ding et al. [30])**.** For all $\theta \in \mathbb{R}^d$, there exists $\gamma > 0$ s.t. the Fisher information matrix $F_\rho(\theta)$ induced by policy $\pi_\theta$ and initial state distribution $\rho$ satisfies

$$F_\rho(\theta) = \mathbb{E}_{x \sim \rho, y \sim \pi_\theta(\cdot|x)} \left[ \nabla_\theta \log \pi_\theta(y|x) \nabla_\theta \log \pi_\theta(y|x)^\top \right] \geqslant \gamma \mathbf{I}_d.$$

This assumption is commonly used in the literature on non-private PG methods [6, 30, 5, 31]. As shown in Sec B.2 in Ding et al. [30], this assumption is satisfied by the Gaussian policy and even certain neural policies.

The next condition is the so-called *compatible function approximation*, which is also a common assumption in the PG literature to handle function approximation error in the non-tabular case.

**Assumption 3** (Compatible, adapted from Assumption 4.6 in Ding et al. [30])**.** For all $\theta \in \mathbb{R}^d$, there exists $\alpha_{\mathsf{bias}} > 0$ such that the *transferred compatible function approximation error* satisfies

$$\mathbb{E}_{x \sim \rho, y \sim \pi_{\theta^*}(\cdot|s)} \left[ (A^{\pi_\theta}(x, y) - u^{*\top} \nabla_\theta \log \pi_\theta(y|x))^2 \right] \leqslant \alpha_{\mathsf{bias}}, \tag{5}$$

where $\pi_{\theta^*}$ is an optimal policy and $u^* = F_\rho(\theta)^\dagger \nabla J(\theta)$.

The "compatible" here means that we are approximating the advantage function $A^{\pi_\theta}(s, a)$ using the $\nabla_\theta \log \pi_\theta(a|s)$ as the feature vector; The "transfer error" here means that we are shifting to the expectation over an optimal policy (rather than the current policy). The error $\alpha_{\mathsf{bias}}$ is zero for a softmax tabular policy and small when $\pi_\theta$ is a rich neural policy. [30–32].

With the above two additional assumptions along with the LS assumption in Assumption 1, we have the following important result, which implies that the objective $J(\theta)$ satisfies the so-called *relaxed weak gradient domination*.

**Lemma 1** (Lemma 4.7 in Ding et al. [30])**.** *If the policy class $\pi_\theta$ satisfies Assumptions 1, 2 and 3, then we have*

$$J^* - J(\theta) \leqslant \frac{G}{\gamma} \|\nabla J(\theta)\| + \sqrt{\alpha_{\mathsf{bias}}}.$$

This lemma essentially allows us to easily translate a guarantee in terms of FOSP to a certain global optimum convergence. This leads to our next main result with its proof given in Appendix E.2.

**Theorem 3.** *Consider the same setting of Theorem 2 and further let Assumptions 2 and 3 hold. Then, for any $\alpha > 0$,* DP-PG *enjoys the following average regret guarantee*

$$J^* - \frac{1}{T} \sum_{t=1}^{T} \mathbb{E}[J(\theta_t)] \leqslant O(\alpha) + O\left(\sqrt{\alpha_{\mathsf{bias}}}\right),$$

*when the sample size satisfies $N \geqslant O_\delta \left( \frac{1}{\alpha^4 \gamma^4} + \frac{\sqrt{d}}{\alpha^3 \gamma^3 \varepsilon} \right)$.*

*Remark* 5. In the above bound, we explicitly include the parameter $\gamma$ to clearly illustrate its impact. The first term $O\left(\frac{1}{\alpha^4 \gamma^4}\right)$ matches the non-private one in Yuan et al. [6] while the second term is the privacy cost. As we can see, for both terms, there exists an additional $1/\gamma$ factor compared to the sample complexity of FOSP. Thus, for very small but still positive $\gamma$, our bound could be large.

Our second scenario is about the specific policy class of softmax in the tabular setting, which allows us to get rid of the parameter $\gamma$. Due to space limit, we relegate these results to Appendix B.

## 6 Differentially Private NPG and REBEL

In this section, we turn to DP-NPG and DP-REBEL, private variants of NPG and RRBEL, and analyze their sample complexities. In particular, we will consider a general private regression oracle as the PrivUpdate in Algorithm 1 and then give concrete examples under different specific regression oracles. Given the similarity, we will mainly focus on DP-NPG in the main paper and relegate the detailed discussion on DP-REBEL to Appendix D.

## 6.1 A Master Algorithm and Guarantee

Our proposed `DP-NPG` is Algorithm 1 with its `PrivUpdate` being instantiated in Algorithm 3 below, which relies on a general private regression oracle to return an approximate minimizer of an estimation problem under the square loss. The square loss in (6) is almost the same as before, as in (2), except that we now take the expectation over a general base policy $\mu$ rather than the specific on-policy $\pi_{\theta_t}$. This update is often called approximate NPG in the literature [5]. As will be shown later, the performance of the algorithm will depend on the choice of $\mu$ in terms of its coverage.

---

**Algorithm 3** `PrivUpdate` Instantiation for `DP-NPG`

---

1: **Input:** $D_t = \{(x_i, y_i, \widehat{A}_t(x_i, y_i))\}_{i=1}^m$, current policy $\theta_t$, base policy $\mu$, learning rate $\eta$, `PrivLS` oracle
2: **Output:** $\theta_{t+1}$
3: Call the `PrivLS` oracle on $D_t := \{(x_i, y_i, \widehat{A}_t(x_i, y_i))\}$ to find an approximate minimizer $w_t$ of

$$\underset{w \in \mathcal{W}}{\operatorname{argmin}} F_t(w) := \mathbb{E}_{x \sim \rho, y \sim \mu(\cdot|x)} \left[ \left( A^{\pi_{\theta_t}}(x,y) - w^\top \nabla \log \pi_{\theta_t}(y|x) \right)^2 \right] \quad (6)$$

4: Output policy $\theta_{t+1} = \theta_t + \eta w_t$

---

We now aim to establish a generic performance guarantee of `DP-NPG`. To start with, we assume that the approximate minimizer $w_t$ returned by `PrivLS` at each iteration satisfies the following guarantee.

**Assumption 4** (Private estimation error). For each $t \in [T]$, the `PrivLS` oracle satisfies $(\varepsilon, \delta)$-DP while ensuring that with probability at least $1 - \zeta$,

$$\mathbb{E}_{x \sim \rho, y \sim \mu(\cdot|x)} \left[ \left( A^{\pi_{\theta_t}}(x,y) - w_t^\top \nabla \log \pi_{\theta_t}(y|x) \right)^2 \right] \leqslant \operatorname{err}_t^2(m, \varepsilon, \delta, \zeta),$$

for some error function $\operatorname{err}_t^2(m, \varepsilon, \delta, \zeta)$ over batch size $m$, privacy parameters $\varepsilon$, $\delta$, and probability $\zeta$.

In addition, we assume standard regularity assumptions commonly used even in the non-private case.

**Assumption 5** ($\beta$-smoothness and boundedness). $\log \pi_\theta(y|x)$ is a $\beta$-smooth function of $\theta$ for all $x, y$, i.e.,

$$\|\nabla_\theta \log \pi_\theta(y|x) - \nabla_{\theta'} \log \pi_{\theta'}(y|x)\|_2 \leqslant \beta \|\theta - \theta'\|_2.$$

Moreover, there exists a constant $W > 0$ such that for all $t \in [T]$, the weight vectors $w_t$ generated by the update rule satisfy $\|w_t\|_2 \leq W$.

Our main result is given by the following theorem.

**Theorem 4** (Master theorem). *Let Assumptions 4 and 5 hold. Then,* `DP-NPG` *satisfies $(\varepsilon, \delta)$-DP as in Definition 2. Moreover, if $\pi_1 := \pi_{\theta_1}$ is a uniform distribution at each state and $\eta = \sqrt{\frac{2 \log |\mathcal{Y}|}{T \beta W^2}}$, with probability at least $1 - \zeta$, for any comparator policy $\pi^*$, we have*

$$J(\pi^*) - \frac{1}{T} \sum_{t=1}^T J(\pi_t) \leqslant \sqrt{\frac{\beta W^2 \log |\mathcal{Y}|}{2T}} + \frac{\sqrt{C_{\mu \to \pi^*}}}{T} \sum_{t=1}^T \operatorname{err}_t(m, \varepsilon, \delta, \zeta),$$

*where $C_{\mu \to \pi^*} := \max_{x,y} \frac{\pi^*(y|x)}{\mu(y|x)}$ and $\pi_t := \pi_{\theta_t}$.*

We use the most intuitive coverage definition for $C_{\mu \to \pi^*}$, i.e., the density ratio, for illustrating the key idea. One can easily extend it to other types of coverage, e.g., relative condition number for the linear case [5] and transfer coefficient for general function classes [33].

## 6.2 Applications

With the above master theorem in hand, we now only need to determine the estimation error under different types of `PrivLS`. To start with, we consider a general `PrivLS` under general function classes for both reward and policy. Specifically, it runs approximate least squares with the exponential

mechanism [34] for privacy protection, as detailed in Algorithm 4 below. To determine its estimation error, we will first present a new result, which could be of independent interest.

---

**Algorithm 4** `PrivLS` Instantiation for `DP-NPG` via Exponential Mechanism

---

1: **Input:** $D_t = \{(x_i, y_i, \widehat{A}_t(x_i, y_i))\}_{i=1}^m$, privacy budget $\varepsilon$, current policy $\theta_t$, reward range $R_{\max}$
2: **Output:** $w_t$
3: Sample $w_t \in \mathcal{W}$ with the following distribution:

$$P(w) \propto \exp\left(-\frac{\varepsilon}{8R_{\max}^2} \cdot L(w)\right) \ \forall w \in \mathcal{W},$$

where $L(w) := \sum_{i \in [m]} [w^\top \nabla \log \pi_{\theta_t}(y_i|x_i) - \widehat{A}_t(x_i, y_i)]^2$

---

**Lemma 2** (Private LS with exponential mechanism). *Let $R > 0$, $\zeta \in (0,1)$, we consider a general sequential estimation setting with an instance space $\mathcal{U}$ and target space $\mathcal{Z}$. Let $\mathcal{H} : \mathcal{U} \to [-R, R]$ be a class of real-valued functions. Let $D = \{(u_i, z_i)\}_{i=1}^m$ be a dataset of $m$ points where $u_i \sim \rho_i = \rho_i(u_{1:i-1}, z_{1:i-1})$, and $z_i = h^*(u_i) + \eta_i$, where $\eta_i$ is zero-mean noise and $h^*$ satisfies approximate realizability, i.e.,*

$$\inf_{h \in \mathcal{H}} \frac{1}{m} \sum_{t=1}^m \mathbb{E}_{u \sim \rho_i}\left[(h^*(u) - h(u))^2\right] \leqslant \alpha_{\mathsf{approx}}. \tag{7}$$

*Suppose $\max_i |z_i| \leqslant R$ and $\max_u |h^*(u)| \leqslant R$. Then, sampling $\widehat{h}$ via the following distribution from exponential mechanism*

$$P(h) \propto \exp\left(-\frac{\varepsilon}{8R^2} \cdot L(h)\right) \ \forall h \in \mathcal{H},$$

*with $L(h) := \sum_{i \in [m]} [h(u_i) - z_i]^2$, yields that*

$$\sum_{i=1}^m \mathbb{E}_{u \sim \rho_i}[(\widehat{h}(u_i) - h^*(u_i))^2] \lesssim R^2 \log(|\mathcal{H}|/\zeta) + R^2 \frac{\log(|\mathcal{H}|/\zeta)}{\varepsilon} + m \cdot \alpha_{\mathsf{approx}}.$$

This lemma can be viewed as the private variant of Lemma 15 in [33]. To leverage this lemma for Algorithm 4, we observe the following mappings for each iteration $t$: $\mathcal{H}$ maps to $\mathcal{W}$, $u_i = (\pi_{\theta_t}, x_i, y_i)$, $h(u_i) = w^\top \nabla \log \pi_{\theta_t}(y_i|x_i)$, $z_i = \widehat{A}_t(x_i, y_i)$ with $\mathbb{E}[z_i] = A^{\pi_{\theta_t}}(x_i, y_i)$, which can be rewritten as an unknown function $w^* = h^*$ over $(\pi_{\theta_t}, x_i, y_i)$. Finally, in our case, $\rho_i$ is non-sequential and fixed during each update, i.e., $x_i \sim \rho, y_i \sim \mu(\cdot|x_i)$ and $\pi_{\theta_t}$ is fixed at $t$. Thus, the approximation error condition in (7) translates to the following one:

$$\inf_{w \in \mathcal{W}} \mathbb{E}_{x \sim \rho, y \sim \mu(\cdot|x)}\left[(A^{\pi_{\theta_t}}(x, y) - w^\top \nabla \log \pi_{\theta_t}(y|x)^2\right] \leqslant \alpha_{\mathsf{approx}}. \tag{8}$$

Based on these discussions, we have the following guarantee of `DP-NPG` with Algorithm 4.

**Corollary 1** (General function class). *Consider* `DP-NPG` *with* `PrivLS` *as in Algorithm 4. Then,* `DP-NPG` *satisfies $(\varepsilon, 0)$-DP. Suppose for each $t \in [T]$, there exists an $\alpha_{\mathsf{approx}}$ such that (8) holds. Then, under the same assumptions in Theorem 4, we have*

$$J(\pi^*) - \frac{1}{T} \sum_{t=1}^T J(\pi_t) \lesssim \sqrt{\frac{\beta W^2 \log |\mathcal{Y}|}{T}} + \sqrt{C_{\mu \to \pi^*} \alpha_{\mathsf{approx}}} + \sqrt{C_{\mu \to \pi^*} \cdot \frac{(1 + 1/\varepsilon) \log(|\mathcal{W}|/\zeta)}{m}}.$$

*This implies that, for a given suboptimality gap of $O(\alpha + \sqrt{C_{\mu \to \pi^*} \alpha_{\mathsf{approx}}})$, the sample complexity bound is $N = T \cdot m = \widetilde{O}\left(\left(\frac{1}{\alpha^4} + \frac{1}{\alpha^4 \varepsilon}\right) \cdot \log|\mathcal{W}| \cdot \beta W^2\right)$.*

*Remark* 6. Several remarks are in order. First, due to the exponential mechanism, we achieve pure DP (i.e., $\delta = 0$) rather than approximate DP; Second, our results hold for general reward and policy function classes. We state the result for a finite $\mathcal{W}$ for ease of presentation. It can be easily extended for an infinite class using a standard covering argument. For instance, if $\mathcal{W} = \mathbb{R}^d$, then $\log(|\mathcal{W}|)$ can

be converted to $\widetilde{O}(d)$. Meanwhile, we remark that in general, Algorithm 4 is not computationally efficient. Thus, the above result is mainly from the statistical perspective. Note that we also explore several efficient oracles for specific scenarios later. Finally, we highlight that the approximation error in (8) is different from the transfer error in (5), which directly takes expectation over the target optimal policy. In contrast, when we use $\alpha_{\mathsf{approx}}$, we have to account for the transition from $\mu$ to $\pi^*$ explicitly via $C_{\mu \to \pi^*}$. This is the case even in the non-private case (cf. Corollary 21 in [5]).

**Log-linear policy class with realizability.** We now turn to computationally efficient `PrivLS` oracles by considering a concrete case where the policy is log-linear and reward $r$ is realizable. More specifically, the policy $\pi_\theta$ is given by some $\theta \in \mathbb{R}^d$ and feature vector $\phi_{x,y} \in \mathbb{R}^d$ with $\pi_\theta(y \mid x) = \frac{\exp(\theta^\top \phi_{x,y})}{\sum_{y' \in \mathcal{Y}} \exp(\theta^\top \phi_{x,y'})}$ and $\|\phi_{x,y}\| \leq B$.

This leads to the fact that $\nabla_\theta \log \pi_\theta(y|x) = \phi_{x,y} - \mathbb{E}_{y' \sim \pi_\theta(\cdot|x)}[\phi_{x,y'}]$ and smoothness parameter of $B^2$. Further, we assume the reward $r(x,y) = \langle w^*, \phi_{x,y} \rangle$ is a linear function with respect to $\phi_{x,y}$, i.e., compatible realizability. In this case, Assumption 4 reduces to estimation error (or in-distribution generalization error) of linear regression:

$$\mathbb{E}_{x \sim \rho, y \sim \mu(\cdot|x)} \left[ \left( \langle w_t - w^*, \bar{\phi}_{x,y}^t \rangle \right)^2 \right] \leqslant \mathrm{err}_t^2(m, \varepsilon, \delta, \zeta), \tag{9}$$

where $\bar{\phi}_{x,y}^t := \phi_{x,y} - \mathbb{E}_{y' \sim \pi_{\theta_t}(\cdot|x)}[\phi_{x,y'}]$, which depends on the current policy.

The above particular form allows us to leverage recent advances in private linear regression, both in the low-dimension and high-dimension cases, respectively.

**Corollary 2** (Log-linear policy in low-dimension). *Consider* `DP-NPG` *with the above log-linear class (with smoothness parameter $\beta = B^2$). Suppose* `PrivLS` *is instantiated with the* ISSP *algorithm in [35]. Then, by [35, Theorem 5], we have that $\mathrm{err}_t(m, \varepsilon, \delta, \zeta) \leqslant \alpha$, when $m \geqslant$* $\widetilde{O}\left( \frac{d}{\alpha^2} + \frac{d\sqrt{\log(1/\delta)}}{\alpha\varepsilon} + \frac{d(\log(1/\delta))^2}{\varepsilon^2} \right)$. *Thus, by Theorem 4, for a suboptimality gap of $O(\alpha)$, the sample complexity bound is $N = T \cdot m = \widetilde{O}_\delta \left( \left( \frac{d}{\alpha^4} + \frac{d}{\alpha^3 \varepsilon} + \frac{d}{\alpha^2 \varepsilon^2} \right) \cdot B^2 W^2 \right)$.*

**Corollary 3** (Log-linear policy in high-dimension). *Consider* `DP-NPG` *with the above log-linear class (with smoothness parameter $\beta = B^2$). Suppose* `PrivLS` *is instantiated with Algorithm 5 in [36]. Then, by [36, Theorem 6.2], we have that $\mathrm{err}_t(m, \varepsilon, \delta, \zeta) \leqslant \alpha$ when $m \geqslant$* $\widetilde{O}\left( \frac{\log(1/\zeta)}{\alpha^4} + \frac{\sqrt{\log(1/\zeta)\log(1/\delta)}}{\alpha^3 \varepsilon} \right)$. *Thus, by Theorem 4, for a suboptimality gap of $O(\alpha)$, the sample complexity bound is $N = T \cdot m = \widetilde{O}_\delta \left( \left( \frac{1}{\alpha^6} + \frac{1}{\alpha^5 \varepsilon} \right) \cdot B^2 W^2 \right)$.*

One key subtlety behind these two corollaries is that $W$ can be large and depend on $d$. This is due to the fact that the update $w_t$ in both `PrivLS` oracles is privatized by a Gaussian noise in the last step. Thus, by standard concentration of a Gaussian vector, $W$ can be on the order of $\sqrt{d}$. This subtlety is somewhat unique due to the interplay between `PrivLS` oracle and "regret-lemma" type analysis in Theorem 4. In practice, one can properly truncate $w_t$, which preserves privacy. In theory, we conjecture that one may use a different technique (e.g., based on the three-point lemma in [37]) to avoid the requirement of bounded $w_t$, which is left to be an exciting future work.

**Connection to private stochastic optimization.** In the context of the above discussion and corollaries, one in-between solution is to aim for suffering dimension dependence only in the private term. This leads us to consider private SGD over a bounded domain, since the estimation error in (9) is equivalent to excess population risk under realizability. Thus, one can leverage existing results on private stochastic optimization to bound $\mathrm{err}_t(m, \varepsilon, \delta, \zeta)$, which does not have dimension dependence in the non-private term, but at a cost of a slower rate $O(1/\sqrt{m})$ vs. $O(1/m)$. See Appendix C.

# 7 Conclusion

We initiate a systematic theoretical investigation into the sample complexity of differentially private PO, leveraging a unified meta-algorithmic framework and reductions to private regression problems. We establish the first set of sample complexity results for several widely used PO algorithms under differential privacy constraints, including `DP-PG`, `DP-NPG` and `DP-REBEL`. Our analysis not only quantifies the privacy cost in PO but also uncovers subtle and important interplays between privacy

mechanisms and algorithmic structures in PO. These insights offer practical guidance for designing privacy-preserving PO methods. We hope our work will open new avenues for future research in both the theoretical understanding and empirical development of private policy optimization. Besides, although we mainly focus on the theory in the main body, we have also managed to conduct some experiments as proof of concept, see Appendix H for details.

## Acknowledgements

This work was supported in part by the National Science Foundation (NSF) under Grant Nos. CAREER-2441519, CNS-2312835, and CNS-2153220.

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

# Contents of Appendix

## A  Additional Related Work

**Policy optimization.** The theoretical study of policy optimization can be roughly divided into two lines of work. The first line often assumes a certain reachability (coverage) condition and takes a perspective from optimization. Under this assumption, various types of policy gradient methods have been investigated, including REINFORCE [1], variance-reduction variants [38], and preconditioned variants such as NPG [11], TRPO [39], and PPO [3]. The performance metrics are often convergence rate or sample complexity, see some typical results in [5, 40, 6, 41, 42]. The second line of work focuses on the exploration setting, i.e., without the coverage condition. To this end, the algorithm design is often based on optimism, e.g., an optimistic version of NPG or PPO via bonus terms or global optimism. Several recent papers have made progress in this direction for tabular RL [43], linear mixture MDP [44], linear MDP and more [45–47]. Our paper can be viewed as the first work that aims to privatize the first line of work above.

**Differentially private RL and bandits.** Recently, a line of work studies RL (bandits) under the constraint of differential privacy, e.g., multi-armed bandits (MABs) [48–52], contextual bandits [27, 36, 53–55], and RL [25, 14, 28, 15, 56]. To the best of our knowledge, only [14] and [15] studied private RL under policy optimization. They consider the exploration setting and characterize the cost of privacy in regret bounds by privatizing optimistic versions of PPO (NPG) under tabular or linear function approximations, respectively. In contrast, we take the optimization perspective (with the coverage condition) and study private PO for general reward/policy function approximations.

**Differentially private stochastic optimization.** Extensive research has been done around private stochastic convex and non-convex optimization. In particular, for the problem of differentially private stochastic convex optimization (DP-SCO), [24] gave the first optimal algorithm in terms of excess population loss, and [57] developed the first linear-time efficient and optimal algorithm. There are also many follow-up papers under various settings, e.g., [58–62]. Moving to the non-convex case, the performance metrics include first-order or second-order population stationary points (e.g., [63–67]) as well as excess population loss (e.g., [66]). As already mentioned, these results in private stochastic optimization can be useful to us since the estimation error in our master theorem is equivalent to excess population loss/risk under square loss and realizability, see Appendix C for more details.

**Differentially private linear regression.** For the specific problem of private linear regression (which is a special case of DP-SCO), one can possibly achieve better results by leveraging its structure. We can roughly group the research efforts[1] in this area based on the conditions on the boundedness of the feature vector (and the true parameter). Assuming boundedness, state-of-the-art results are established in [68, 69]. On the other hand, without boundedness but under a general sub-Gaussian data, ISSP in [35] is the first efficient and nearly-optimal algorithm, which does not require an identity covariance matrix compared with [70, 71] and does not depend on the condition number of the covariance matrix compared with [72, 73]. Moving from approximate DP to pure DP, the very recent work [74] gives the first polynomial-time and sample-optimal private regression algorithm. As mentioned before, these results on private linear regression are useful to us since, under log-linear policy parameterization, our estimation error reduces to the estimation error in linear regression.

## B  Tabular Softmax with Log-barrier Regularization

In this section, we move to the second scenario for our DP-PG of global convergence where we consider the tabular case with the classic softmax policy:

**Definition 3** (Tabular softmax policy). Consider a finite state space $\mathcal{X}$ and action space $\mathcal{Y}$. For any state-action pair $(x, y) \in \mathcal{X} \times \mathcal{Y}$, the softmax policy is given by

$$\pi_\theta(y|x) = \frac{\exp(\theta_{x,y})}{\sum_{y' \in \mathcal{Y}} \exp(\theta_{x,y'})},$$

where $\theta \in \mathbb{R}^{|\mathcal{X}||\mathcal{Y}|}$.

One key motivation here is to leverage the tabular structure and the specific property of softmax policy to establish a sample complexity of global optimum convergence that is independent of the parameter $\gamma$. To this end, as in the non-private case [5, 6], we will consider a regularized problem, whose FOSP turns out to be an approximate global optimal solution of the unregularized (original) objective, for proper choice of regularization. In particular, we consider the following log-barrier regularization objective:

$$J_\lambda(\theta) := J(\theta) - \lambda \mathbb{E}_{x \sim \text{Unif}_\mathcal{X}} \left[ \text{KL}(\text{Unif}_\mathcal{Y}, \pi_\theta(\cdot|x)) \right]$$

$$= J(\theta) + \frac{\lambda}{|\mathcal{Y}||\mathcal{X}|} \sum_{x,y} \log \pi_\theta(y|x) + \lambda \log |\mathcal{Y}|, \tag{10}$$

where the KL divergence is $\text{KL}(p, q) = \mathbb{E}_{x \sim p} \left[ \log \frac{p(x)}{q(x)} \right]$, $\text{Unif}_\chi$ denotes the uniform distribution over a set $\chi$ and $\lambda > 0$ is the regularization constant.

We will run our DP-PG over this regularized objective by using the sample-based gradient estimator at each step with proper choices of batch size and learning rate. Then, we have the following main result regarding the global optimum convergence in terms of the unregularized $J(\theta)$. The proof is given in Appendix E.3.

**Theorem 5.** *Consider Algorithm 2 applied to $J_\lambda(\theta)$. For any $m > 0$, setting $\sigma^2 = \frac{16 \ln(1.25/\delta) \cdot R_{\max}^2 G^2}{m^2 \varepsilon^2}$ ensures $(\varepsilon, \delta)$-DP as in Definition 2. Further, there exist proper choices of parameters for $m$ and $\eta$, such that the following holds*

$$J^* - \frac{1}{T} \sum_{t=1}^{T} \mathbb{E}\left[J(\theta_t)\right] \leqslant O(\alpha),$$

---

[1] As before, this is not a complete list of all the works in this area.

*when the sample size satisfies* $N \geqslant O\left(\frac{1}{\alpha^6} + \frac{\sqrt{d}}{\alpha^{9/2}\varepsilon}\right)$.

*Remark* 7. The first term in the sample complexity bound matches the non-private one in Yuan et al. [6], while the second term is the lower-order privacy cost (for constant $\varepsilon$ and $d$). We note that while the dependence on $\alpha$ is worse than the previous one, there is no dependence on $\gamma$ in the bound, which could offer benefits when $\gamma$ is quite small.

## C   Connection to Private Stochastic Optimization

In this section, we first aim to bound the estimation error in (9) by leveraging the existing result in private SCO. In particular, we aim to apply Theorem 3.2 in [24] for a Lipschitz and smooth loss function. The first step is to realize that under realizability, the LHS in (9) is equal to the excess population loss for a square loss, which is both Lipschitz (with parameter of order $O(B^2W)$ under our boundedness assumption for both feature and parameter space) and smooth (with parameter of $O(B^2)$). Hence, by [24, Theorem 3.2], we can obtain that $\mathrm{err}_t(m, \varepsilon, \delta, \zeta) \leqslant \alpha$ when $m \geqslant \widetilde{O}\left(\frac{1}{\alpha^4} + \frac{\sqrt{d\log(1/\delta)}}{\alpha^2\varepsilon}\right) \cdot O(\mathsf{Poly}(B, W))$. Thus, by Theorem 4, for a suboptimality gap of $O(\alpha)$, the sample complexity bound is $N = T \cdot m = \widetilde{O}_\delta\left((\frac{1}{\alpha^6} + \frac{\sqrt{d}}{\alpha^4\varepsilon}) \cdot \mathsf{Poly}(B, W)\right)$. Recall that due to projection in the mini-batch SGD in [24], one can ensure that $\|w_t\| \leq W$ for some $W$ that is independent of $d$. Thus, the non-private term does not depend on $d$. We mention in passing that one can also potentially directly bound the error in Assumption 4 by leveraging the excess population loss for non-convex loss functions.

## D   Differentially Private REBEL

In this section, we will consider DP-REBEL which includes a general private regression oracle as the PrivUpdate in Algorithm 1 and then similar to our previous DP-NPG, we will give some concrete applications under different specific regression oracles.

Our proposed DP-REBEL is Algorithm 1 with its PrivUpdate being instantiated in Algorithm 5, which also relies on a general private least-square regression oracle to return an approximate minimizer of an estimation problem under the square loss. The square loss in (11) is the same as before in (3).

---

**Algorithm 5** PrivUpdate Instantiation for DP-REBEL

---

1: **Input:** $D_t = \{(x_i, y_i, \widehat{A}_t(x_i, y_i))\}_{i=1}^m$, current policy $\theta_t$, base policy $\mu$, learning rate $\eta$, PrivLS oracle
2: **Output:** $\theta_{t+1}$
3: Call the PrivLS oracle on $D_t := \{(x_i, y_i, \widehat{A}_t(x_i, y_i))\}$ to find an approximate minimizer $w_t$ of

$$\operatorname*{argmin}_{\theta \in \Theta} F_t(\theta) = E\left[\frac{1}{\eta}\left(\ln\frac{\pi_\theta(y|x)}{\pi_{\theta_t}(y|x)} - \ln\frac{\pi_\theta(y'|x)}{\pi_{\theta_t}(y'|x)}\right) - \left(\widehat{A}_t(x_i, y_i)\right)\right]^2, \qquad (11)$$

where expectation is over $x \sim \rho, y \sim \mu(\cdot|x), y' \sim \pi_{\theta_t}(\cdot|x)$.
4: Output policy $\theta_{t+1} = \theta_t + \eta w_t$

---

We now turn to establish a generic performance guarantee of DP-REBEL. Similar to DP-NPG, we assume that the approximate minimizer $w_t$ returned by PrivLS at each iteration satisfies the following guarantee.

**Assumption 6** (Private estimation error). For each $t \in [T]$, the PrivLS oracle satisfies $(\varepsilon, \delta)$-differential privacy and ensures that with probability at least $1 - \zeta$,

$$\mathbb{E}\left[\frac{1}{\eta}\left(\ln\frac{\pi_{\theta_{t+1}}(y|x)}{\pi_{\theta_t}(y|x)} - \ln\frac{\pi_{\theta_{t+1}}(y'|x)}{\pi_{\theta_t}(y'|x)}\right) - \left(\widehat{A}_t(x_i, y_i)\right)\right]^2 \leqslant \mathrm{err}_t^2(m, \varepsilon, \delta, \zeta),$$

for some statistical error function $\mathrm{err}_t^2(m, \varepsilon, \delta, \zeta)$ depending on batch size $m$, privacy parameters $(\varepsilon, \delta)$, and probability $\zeta$, also, expectation here is over $x \sim \rho, y \sim \mu(\cdot|x), y' \sim \pi_{\theta_t}(\cdot|x)$.

*Remark* 8. This assumption parallels Assumption 4 in `DP-NPG` and ensures that our private oracle accurately estimates the relative reward differences while preserving our DP in PO as Definition 2.

Our main result is given by the following theorem.

**Theorem 6.** *Under Assumption 6, according to Lemma 11 and Lemma 12, we have with probability at least $1 - \zeta$, for any comparator policy $\pi^*$ such that:*

$$J(\pi^*) - \frac{1}{T} \sum_{t=1}^{T} J(\pi_t) \leqslant 2A\sqrt{\frac{\ln |\mathcal{Y}|}{T}} + \frac{\sqrt{10 C_{\mu \to \pi^*}}}{T} \sum_{t=1}^{T} \mathrm{err}_t(m, \varepsilon, \delta, \zeta).$$

*Remark* 9. If we simply set the base policy $\mu = \pi_{\theta_t}$, which make this assumption simpler, then we can have a tighter bound, it is easy to show that the bound will turns to $2A\sqrt{\frac{\ln |\mathcal{Y}|}{T}} + \frac{\sqrt{2 C_{\mu \to \pi^*}}}{T} \sum_{t=1}^{T} \mathrm{err}_t(m, \varepsilon, \delta, \zeta)$.

The application here is totally same as the `DP-NPG`, thus we can directly use our previous Corollary 1, Corollary 2 and Corollary 3 which derive the almost same bound of sample complexity. For `PrivLS` with exponential mechanism, consider `DP-REBEL` with `PrivLS` as in Algorithm 4, for a given suboptimality gap of $O(\alpha + \sqrt{C_{\mu \to \pi^*} \alpha_{\mathsf{approx}}})$, the sample complexity bound is $N = T \cdot m = \widetilde{O}\left((\frac{1}{\alpha^4} + \frac{1}{\alpha^4 \varepsilon}) \cdot \log |\mathcal{W}| \cdot A^2\right)$. For log-linear policy class with realizability, assume that $\mathrm{err}_t(m, \varepsilon, \delta, \zeta) \leqslant \alpha$, therefore, for log-liner-policy in low-dimension and high-dimension, we have $m \geqslant \widetilde{O}\left(\frac{d}{\alpha^2} + \frac{d\sqrt{\log(1/\delta)}}{\alpha \varepsilon} + \frac{d(\log(1/\delta))^2}{\varepsilon^2}\right)$, and $m \geqslant \widetilde{O}\left(\frac{\log(1/\zeta)}{\alpha^4} + \frac{\sqrt{\log(1/\zeta) \log(1/\delta)}}{\alpha^3 \varepsilon}\right)$, respectively. Thus, we can derive such sample complexity: $N = T \cdot m = \widetilde{O}_\delta\left((\frac{d}{\alpha^4} + \frac{d}{\alpha^3 \varepsilon} + \frac{d}{\alpha^2 \varepsilon^2}) \cdot A^2\right)$ for log-liner-policy in low-dimension and $N = T \cdot m = \widetilde{O}_\delta\left((\frac{1}{\alpha^6} + \frac{1}{\alpha^5 \varepsilon}) \cdot A^2\right)$ for high-dimension.

# E    Proof of Chapter 5

## E.1    Proof of Theorem 2

**Lemma 3** (ABC)**.** *There exists constants $A, B, C \geqslant 0$ such that the policy gradient estimator satisfies:*

$$\mathbb{E}\left[\left\|\widetilde{\nabla}_m J(\theta)\right\|^2\right] \leqslant 2A(J^* - J(\theta)) + B \|\nabla J(\theta)\|^2 + C, \tag{12}$$

*where $\nabla J(\theta) = \mathbb{E}_{x \sim \rho, y \sim \pi_\theta(\cdot|s)}[A^{\pi_\theta}(x, y)\nabla_\theta \log \pi_\theta(y|x)]$, and $A = 0, B = 1 - 1/m, C = \frac{4 R_{\max}^2 G^2}{m} + d\sigma^2$.*

*Proof.* For notation simplicity, we let $g_\theta(\tau_i) := A^{\pi_\theta}(x_i, y_i)\nabla_\theta \log \pi_\theta(y_i|x_i)$. Thus, we have $\widetilde{\nabla}_m J(\theta) = \frac{1}{m} \sum_i g_\theta(\tau_i) + Z$. Notice that $\mathbb{E}[g_\theta(\tau_i)] = \mathbb{E}\left[\widetilde{\nabla}_m J(\theta)\right] = \nabla J(\theta)$, cause $Z$ is the gaussian bias, which expectation is 0.

Now, we have

$$
\mathbb{E}\left[\left\|\widetilde{\nabla}_m J(\theta)\right\|^2\right] = \mathbb{E}\left[\left\|\frac{1}{m}\sum_i g_\theta(\tau_i) + Z\right\|^2\right]
$$

$$
= \mathbb{E}\left[\left\|\frac{1}{m}\sum_i g_\theta(\tau_i)\right\|^2\right] + \mathbb{E}\left[\|Z\|^2\right] + 2\cdot\mathbb{E}\left[\left\langle\frac{1}{m}\sum_i g_\theta(\tau_i), Z\right\rangle\right]
$$

$$
= \mathbb{E}\left[\left\|\frac{1}{m}\sum_i g_\theta(\tau_i)\right\|^2\right] + d\sigma^2 + 0
$$

$$
= \mathbb{E}\left[\left\|\frac{1}{m}\sum_i g_\theta(\tau_i) - \nabla J(\theta) + \nabla J(\theta)\right\|^2\right] + d\sigma^2
$$

$$
= \|\nabla J(\theta)\|^2 + \mathbb{E}\left[\left\|\frac{1}{m}\sum_i g_\theta(\tau_i) - \nabla J(\theta)\right\|^2\right] + d\sigma^2
$$

$$
= \|\nabla J(\theta)\|^2 + \frac{1}{m^2}\sum_i \mathbb{E}\left[\|g_\theta(\tau_i) - \nabla J(\theta)\|^2\right] + d\sigma^2
$$

$$
= \|\nabla J(\theta)\|^2 + \frac{1}{m}\cdot\mathbb{E}\left[\|g_\theta(\tau_1)\|^2 - \|\nabla J(\theta)\|^2\right] + d\sigma^2.
$$

To proceed, we need to establish an upper bound on $\mathbb{E}\left[\|g_\theta(\tau_1)\|^2\right]$. In particular, we have

$$
\mathbb{E}\left[\|g_\theta(\tau_1)\|^2\right] = \mathbb{E}\left[|A^{\pi_\theta}(x_1, y_1)|^2\|\nabla_\theta\log\pi_\theta(y_1\mid x_1)\|^2\right]
$$
$$
\leqslant 4R_{\max}^2 G^2,
$$

which follows from Assumption 1.

Hence, we conclude that:

$$
\mathbb{E}\left[\left\|\widetilde{\nabla}_m J(\theta)\right\|^2\right] \leqslant \left(1 - \frac{1}{m}\right)\|\nabla J(\theta)\|^2 + \frac{4R_{\max}^2 G^2}{m} + d\sigma^2.
$$

i.e., ABC condition in (12) is satisfied with $A = 0, B = 1 - 1/m, C = \frac{4R_{\max}^2 G^2}{m} + d\sigma^2$ $\qquad\square$

**Lemma 4** (Smoothness under LS). *Under LS assumption in Assumption 1, $J(\cdot)$ is L-smooth, namely $\left\|\nabla^2 J(\theta)\right\| \leqslant L$ for all $\theta$, with*

$$
L = 2R_{\max}(G^2 + F).
$$

*Proof.* For smoothness, it suffices to bound the operator norm of Hessian, i.e., $\left\|\nabla^2 J(\theta)\right\|$.

By definition, we have

$$
\nabla^2 J(\theta) = \nabla_\theta\mathbb{E}_{x\sim\rho, y\sim\pi_\theta(\cdot|x)}\left[A^{\pi_\theta}(x, y)\nabla_\theta\log\pi_\theta(y|x)\right]
$$
$$
\overset{(a)}{=} \nabla_\theta\int p_\theta(x, y)\left(A^{\pi_\theta}(x, y)\nabla_\theta\log\pi_\theta(y|x)\right)\,d(x, y)
$$
$$
\overset{(b)}{=} \int \nabla_\theta p_\theta(x, y)\left(A^{\pi_\theta}(x, y)\nabla_\theta\log\pi_\theta(y|x)\right)^\top d(x, y) + \int p_\theta(x, y)\left(A^{\pi_\theta}(x, y)\nabla_\theta^2\log\pi_\theta(y|x)\right)\,d(x, y)
$$
$$
= \mathbb{E}_{x, y\sim p_\theta}\left[A^{\pi_\theta}(x, y)\nabla_\theta\log\pi_\theta(y|x)\log\pi_\theta(y|x)^\top\right] + \mathbb{E}_{x, y\sim p_\theta}\left[A^{\pi_\theta}(x, y)\nabla_\theta^2\log\pi_\theta(y|x)\right]
$$

where in $(a)$, we let $p_\theta(x, y) := \rho(x)\pi_\theta(y|x)$, and $(b)$ holds by chain rules.

Thus, we have

$$\left\|\nabla_\theta^2 J(\theta)\right\| \leq \underbrace{\mathbb{E}_{x,y}\left[|A^{\pi_\theta}(x,y)| \left\|\nabla_\theta \log \pi_\theta(y|x)\right\|^2\right]}_{\mathcal{T}_1} + \underbrace{\mathbb{E}_{x,y}\left[|A^{\pi_\theta}(x,y)| \left\|\nabla_\theta^2 \log \pi_\theta(y|x)\right\|\right]}_{\mathcal{T}_2}.$$

For $\mathcal{T}_1$ and $\mathcal{T}_2$, by Assumption 1, we have

$$\mathcal{T}_1 \leq 2R_{\max}G^2, \quad \mathcal{T}_2 \leq 2R_{\max}F,$$

which hence completes the proof. $\square$

**Lemma 5** (Adapted from Theorem 3.4 in Yuan et al. [6]). *Suppose that $J$ satisfies smoothness and the ABC assumption in Lemma 3. Consider the iterates $\theta_t$ of the PG method with step size $\eta_t = \eta \in (0, \frac{2}{LB})$, let $\delta_1 = J^* - J(\theta_1)$. In particular, if $A = 0$, we have:*

$$E\left[\|\nabla J(\theta_U)\|^2\right] \leqslant \frac{2\delta_1}{\eta T(2 - LB\eta)} + \frac{LC\eta}{2 - LB\eta}. \tag{13}$$

*where $\theta_U$ is uniformly sampled from $\{\theta_1, ..., \theta_T\}$*

*Proof of Theorem 2.* Followed by Lemma 5, when $\eta < \frac{1}{LB}$, we can simplify the equation into this:

$$E\left[\|\nabla J(\theta_U)\|^2\right] \leqslant \frac{2\delta_1}{\eta T} + LC\eta, \tag{14}$$

where $B = 1 - 1/m$, $\delta_1 = J^* - J(\theta_1)$, $L = 2R_{\max}(G^2 + F)$, $C = \frac{4R_{\max}^2 G^2}{m} + d\sigma^2$, $G$ and $F$ are constants.

From Theorem 1, to make sure our algorithm satisfy the $(\varepsilon, \delta)$-DP as in Definition 2, we set $\sigma^2 = \frac{16 \ln(1.25/\delta) \cdot R_{\max}^2 G^2}{m^2 \varepsilon^2}$.

Based on Lemma 5 and Equation (14), choose $\eta = \min\{\frac{1}{LB}, \frac{\sqrt{2\delta_1}}{\sqrt{TLC}}\}$, we have:

$$\begin{aligned}
\mathbb{E}\left[\|\nabla J(\theta_U)\|^2\right] &\leqslant \frac{2\delta_1 LB}{T} + \frac{2\sqrt{2\delta_1 LC}}{\sqrt{T}} \\
&= O\left(\frac{1}{T} + \frac{\sqrt{C}}{\sqrt{T}}\right) \\
&= O\left(\frac{m}{N} + \frac{1}{\sqrt{N}} + \frac{\sigma\sqrt{md}}{\sqrt{N}}\right) \\
&= O\left(\frac{m}{N} + \frac{1}{\sqrt{N}} + \frac{\sqrt{d}}{\varepsilon\sqrt{Nm}}\right).
\end{aligned}$$

To proceed, we need to determine the value of m.

In order to balance the terms in the convergence bound $O\left(\frac{m}{N} + \frac{1}{\sqrt{N}} + \frac{\sqrt{d}}{\varepsilon\sqrt{Nm}}\right)$, we set $\frac{m}{N} = \frac{\sqrt{d}}{\varepsilon\sqrt{Nm}}$.

Thus, we have:

$$m = \left(\frac{\sqrt{d}}{\varepsilon}\right)^{2/3} N^{1/3} = (1/\varepsilon)^{2/3} (Nd)^{1/3}.$$

Substituting back, the convergence bound simplifies to:

$$O\left(\frac{1}{\sqrt{N}} + \left(\frac{\sqrt{d}}{N\varepsilon}\right)^{2/3}\right).$$

$\square$

## E.2 Proof of Theorem 3

*Proof.* We know that:

$$E\left[\|\nabla J(\theta_U)\|^2\right] = \frac{1}{T}\sum_{t=1}^{T}\mathbb{E}\left[\|\nabla J(\theta_t)\|^2\right]. \tag{15}$$

Besides, followed by Lemma 1, we obtain that:

$$(J^* - J(\theta))^2 \leqslant \left(\frac{G}{\gamma}\|\nabla J(\theta)\| + \sqrt{\alpha_{\text{bias}}}\right)^2 \leqslant 2\frac{G^2}{\gamma^2}\|\nabla J(\theta)\|^2 + 2\alpha_{\text{bias}},$$

which holds by $(p+q)^2 \leqslant 2p^2 + 2q^2$.

Taking expectation over both sides, condition on $\theta_t$, yields that

$$\frac{1}{T}\sum_{t=1}^{T}E\left[(J^* - J(\theta))^2\right] \leqslant 2\frac{G^2}{\gamma^2}\frac{1}{T}\sum_{t=1}^{T}\mathbb{E}\left[\|\nabla J(\theta_t)\|^2\right] + 2\alpha_{\text{bias}}$$

$$\stackrel{(15)}{=} 2\frac{G^2}{\gamma^2}E\left[\|\nabla J(\theta_U)\|^2\right] + 2\alpha_{\text{bias}}$$

$$\stackrel{(a)}{=} O\left(\frac{1}{\gamma^2}\left(\frac{1}{\sqrt{N}} + \left(\frac{\sqrt{d}}{N\varepsilon}\right)^{2/3}\right)\right) + O(\alpha_{\text{bias}}),$$

where $(a)$ holds by Theorem 2.

By applying Jensen inequality twice, we have:

$$\frac{1}{T}\sum_{t=1}^{T}E\left[(J^* - J(\theta))^2\right] \geqslant E\left[\left(J^* - \frac{1}{T}\sum_{t=1}^{T}J(\theta_t)\right)^2\right] \geqslant \left(J^* - \frac{1}{T}\sum_{t=1}^{T}E\left[J(\theta_t)\right]\right)^2.$$

So we can derive that:

$$\left(J^* - \frac{1}{T}\sum_{t=1}^{T}E\left[J(\theta_t)\right]\right)^2 \leqslant O\left(\frac{1}{\gamma^2}\left(\frac{1}{\sqrt{N}} + \left(\frac{\sqrt{d}}{N\varepsilon}\right)^{2/3}\right)\right) + O(\alpha_{\text{bias}}).$$

In that case, we finally get the result:

$$J^* - \frac{1}{T}\sum_{t=1}^{T}\mathbb{E}\left[J(\theta_t)\right] = O\left(\frac{1}{\gamma}\left(N^{-1/4} + \left(\frac{\sqrt{d}}{N\varepsilon}\right)^{1/3}\right)\right) + O(\sqrt{\alpha_{\text{bias}}}).$$

Suppose $J^* - \frac{1}{T}\sum_{t=1}^{T}\mathbb{E}\left[J(\theta_t)\right] \leqslant O(\alpha) + O(\sqrt{\alpha_{\text{bias}}})$, we have:

$$N \geqslant O\left(\frac{1}{\alpha^4\gamma^4} + \frac{\sqrt{d}}{\alpha^3\gamma^3\varepsilon}\right).$$

$\square$

## E.3 Proof of Theorem 5

Based on softmax settings in Definition 3, by simple calculus, we have

$$\frac{\partial \log \pi_\theta(y|x)}{\partial \theta_x} = \mathbf{1}_y - \pi_x(\theta), \tag{16}$$

$$\frac{\partial^2 \log \pi_\theta(y|x)}{\partial \theta_x^2} = -\mathbf{H}(\pi_x(\theta)),$$

where $\mathbf{1}_y \in \mathbb{R}^{|\mathcal{Y}|}$ is a vector with all zero entries except being 1 for the entry corresponding to action $y$, and $\mathbf{H}(\pi_x(\theta)) = \mathrm{Diag}(\pi_x(\theta)) - \pi_x(\theta)\pi_x(\theta)^\top$.

In particular, for softmax, we can determine the $G$ and $F$ in Assumption 1:

$$\|\nabla_\theta \log \pi_\theta(y \mid x)\| \leqslant G := \sqrt{1 - \frac{1}{|\mathcal{Y}|}}$$

$$\|\nabla_\theta^2 \log \pi_\theta(y \mid x)\| \leqslant F := 1.$$

### E.3.1   FOSP of softmax policy

**Lemma 6.** *The regularized gradient estimator* $\widetilde{\nabla}_m J_\lambda(\theta)$ *satisfies ABC assumption in Lemma 3 with parameters:*

$$A = 0, \quad B = 1 - \frac{1}{m}$$
$$C = \frac{2}{m}\left(1 - \frac{1}{|\mathcal{Y}|}\right)\left(4R_{\max}^2 + \frac{\lambda^2}{|\mathcal{X}|}\right) + d\sigma^2,$$

*Specifically, we have the variance bound:* $\mathbb{E}\left[\left\|\widetilde{\nabla}_m J_\lambda(\theta)\right\|^2\right] \leqslant \left(1 - \frac{1}{m}\right)\|\nabla J_\lambda(\theta)\|^2 + d\sigma^2 + \frac{2}{m}\left(1 - \frac{1}{|\mathcal{Y}|}\right)\left(4R_{\max}^2 + \frac{\lambda^2}{|\mathcal{X}|}\right).$

*Proof.* Similar to Appendix E.1, here we let $g_\theta(\tau)$ be a stochastic gradient estimator of one single sampled trajectory $\tau$. Thus we have: $\widetilde{\nabla}_m J(\theta) = \frac{1}{m}\sum_i g_\theta(\tau_i) + Z$.

From equation (10) we have the following gradient estimator

$$\widetilde{\nabla}_m J_\lambda(\theta) = \widetilde{\nabla}_m J(\theta) + \frac{\lambda}{|\mathcal{Y}||\mathcal{X}|}\sum_{x,y}\nabla_\theta \log \pi_{x,y}(\theta).$$

For a state $x \in \mathcal{X}$, we have

$$
\frac{\lambda}{|\mathcal{Y}||\mathcal{X}|}\sum_{y\in\mathcal{Y}}\frac{\partial \log \pi_{x,y}(\theta)}{\partial \theta_x} \overset{(16)}{=} \frac{\lambda}{|\mathcal{Y}||\mathcal{X}|}\sum_{y\in\mathcal{Y}}(\mathbf{1}_y - \pi_x(\theta))
$$
$$
= \frac{\lambda \mathbf{1}_{|\mathcal{Y}|}}{|\mathcal{Y}||\mathcal{X}|} - \frac{\lambda}{|\mathcal{X}|}\pi_x(\theta)
$$
$$
= \frac{\lambda}{|\mathcal{X}|}\left(\frac{\mathbf{1}_{|\mathcal{Y}|}}{|\mathcal{Y}|} - \pi_x(\theta)\right),
$$

where $\mathbf{1}_{|\mathcal{Y}|} \in \mathbb{R}^{|\mathcal{Y}|}$ is a vector of all ones.

Thus we have

$$\widetilde{\nabla}_m J_\lambda(\theta) = \widetilde{\nabla}_m J(\theta) + \frac{\lambda}{|\mathcal{X}|}\left(\frac{\mathbf{1}}{|\mathcal{Y}|} - [\pi_x(\theta)]_{x\in\mathcal{X}}\right), \tag{17}$$

where $\mathbf{1} \in \mathbb{R}^{|\mathcal{X}||\mathcal{Y}|}$ and $[\pi_x(\theta)]_{x\in\mathcal{X}} = [\pi_{x_1}(\theta); ...; \pi_{x_{|\mathcal{X}|}}(\theta)] \in \mathbb{R}^{|\mathcal{X}||\mathcal{Y}|}$ is the stacking of the vectors $\pi_x(\theta)$.

Next, taking expectation on the trajectories, we have

$$\mathbb{E}\left[\left\|\widetilde{\nabla}_m J_\lambda(\theta)\right\|^2\right] \overset{(17)}{=} \mathbb{E}\left[\left\|\widetilde{\nabla}_m J(\theta) + \frac{\lambda}{|\mathcal{X}|}\left(\frac{1}{|\mathcal{Y}|} - [\pi_x(\theta)]_{x\in\mathcal{X}}\right)\right\|^2\right]$$

$$= \mathbb{E}\left[\left\|\nabla J(\theta) + \frac{\lambda}{|\mathcal{X}|}\left(\frac{1}{|\mathcal{Y}|} - [\pi_x(\theta)]_{x\in\mathcal{X}}\right) + \widetilde{\nabla}_m J(\theta) - \nabla J(\theta)\right\|^2\right]$$

$$\overset{(a)}{=} \|\nabla J_\lambda(\theta)\|^2 + \mathbb{E}\left[\left\|\widetilde{\nabla}_m J(\theta) - \nabla J(\theta)\right\|^2\right]$$

$$\overset{(b)}{=} \|\nabla J_\lambda(\theta)\|^2 + \mathbb{E}\left[\left\|\widehat{\nabla}_m J(\theta) + \mathbf{Z} - \nabla J(\theta)\right\|^2\right]$$

$$= \|\nabla J_\lambda(\theta)\|^2 + \frac{\mathbb{E}\left[\|g_\theta(\tau_1) - \nabla J(\theta)\|^2\right]}{m} + d\sigma^2$$

$$= \|\nabla J_\lambda(\theta)\|^2 + d\sigma^2$$

$$+ \frac{\mathbb{E}\left[\left\|g_\theta(\tau_1) + \frac{\lambda}{|\mathcal{X}|}\left(\frac{1}{|\mathcal{Y}|} - [\pi_x(\theta)]_{x\in\mathcal{X}}\right) - \nabla J(\theta) - \frac{\lambda}{|\mathcal{X}|}\left(\frac{1}{|\mathcal{Y}|} - [\pi_x(\theta)]_{x\in\mathcal{X}}\right)\right\|^2\right]}{m}$$

$$\overset{(c)}{=} \left(1 - \frac{1}{m}\right)\|\nabla J_\lambda(\theta)\|^2 + \frac{\mathbb{E}\left[\left\|g_\theta(\tau_1) + \frac{\lambda}{|\mathcal{X}|}\left(\frac{1}{|\mathcal{Y}|} - [\pi_x(\theta)]_{x\in\mathcal{X}}\right)\right\|^2\right]}{m} + d\sigma^2$$

$$\overset{(d)}{\leqslant} \left(1 - \frac{1}{m}\right)\|\nabla J_\lambda(\theta)\|^2 + \frac{2\mathbb{E}\left[\|g_\theta(\tau_1)\|^2\right] + 2\left\|\frac{\lambda}{|\mathcal{X}|}\left(\frac{1}{|\mathcal{Y}|} - [\pi_x(\theta)]_{x\in\mathcal{X}}\right)\right\|^2}{m} + d\sigma^2,$$

where $(a)$ and $(c)$ hold by definition of $\nabla J_\lambda(\theta)$; $(b)$ holds by definition of $\widetilde{\nabla}_m J(\theta)$; $(d)$ holds by $(p+q)^2 \leqslant 2p^2 + 2q^2$.

In particular, we have

$$\left\|\frac{\lambda}{|\mathcal{X}|}\left(\frac{1}{|\mathcal{Y}|} - [\pi_x(\theta)]_{x\in\mathcal{X}}\right)\right\|^2 \leqslant \frac{\lambda^2}{|\mathcal{X}|^2}\left(\frac{|\mathcal{X}||\mathcal{Y}|}{|\mathcal{Y}|^2} - 2\frac{|\mathcal{X}|}{|\mathcal{Y}|} + |\mathcal{X}|\right) = \frac{\lambda^2}{|\mathcal{X}|}\left(1 - \frac{1}{|\mathcal{Y}|}\right),$$

where the inequality is obtained by using $\|\pi_x(\theta)\|^2 \leqslant 1$.

As for $\mathbb{E}\left[\|g_\theta(\tau_1)\|^2\right]$, we have

$$\mathbb{E}\left[\|g_\theta(\tau_1)\|^2\right] \leqslant 4R_{\max}^2 G^2 = 4R_{\max}^2\left(1 - \frac{1}{|\mathcal{Y}|}\right),$$

where the equality is obtained by Assumption 1 with $G^2 = \left(1 - \frac{1}{|\mathcal{Y}|}\right)$.

Combining above, we proved the gradient estimator $\widetilde{\nabla}_m J_\lambda(\theta)$ satisfies ABC assumption with

$$\mathbb{E}\left[\left\|\widetilde{\nabla}_m J_\lambda(\theta)\right\|^2\right] \leqslant \left(1 - \frac{1}{m}\right)\|\nabla J_\lambda(\theta)\|^2 + \frac{2}{m}\left(1 - \frac{1}{|\mathcal{Y}|}\right)\left(4R_{\max}^2 + \frac{\lambda^2}{|\mathcal{X}|}\right) + d\sigma^2,$$

where

$$A = 0, \quad B = 1 - \frac{1}{m}, \quad C = \frac{2}{m}\left(1 - \frac{1}{|\mathcal{Y}|}\right)\left(4R_{\max}^2 + \frac{\lambda^2}{|\mathcal{X}|}\right) + d\sigma^2.$$

$\square$

**Lemma 7** (Regularized FOSP Convergence). *Under the learning rate condition $\eta < \frac{1}{LB}$, the iterates satisfy:*

$$\mathbb{E}\left[\|\nabla J_\lambda(\theta_U)\|^2\right] \leqslant \frac{2\delta_1}{\eta T} + LC\eta, \tag{18}$$

*where $B = 1 - 1/m$, $\delta_1 = J^* - J(\theta_1)$, $L = 2R_{\max}(2 - \frac{1}{|\mathcal{Y}|})$, and $C$ as defined in Lemma 6.*

*Proof.* To proceed with the analysis, we first introduce the following key lemma:

**Lemma 8** (Adapted from Lemma E.3 in Yuan et al. [6]). *$J(\cdot)$ with the softmax tabular policy is $2R_{\max}\left(2 - \frac{1}{|\mathcal{Y}|}\right)$-smooth and $2R_{\max}\sqrt{1 - \frac{1}{|\mathcal{Y}|}}$-Lipschitz.*

From Lemma 8, we know that $J_\lambda(\cdot)$ is smooth and Lipschitz. Then, based on Lemma 5, we have:

$$\mathbb{E}\left[\|\nabla J_\lambda(\theta_U)\|^2\right] \leqslant \frac{2\delta_1}{\eta T(2 - LB\eta)} + \frac{LC\eta}{2 - LB\eta}.$$

Assuming $\eta < \frac{1}{LB}$, the above equation simplifies to:

$$\mathbb{E}\left[\|\nabla J_\lambda(\theta_U)\|^2\right] \leqslant \frac{2\delta_1}{\eta T} + LC\eta,$$

where $B = 1 - 1/m$, $\delta_1 = J^* - J(\theta_1)$, $L = 2R_{\max}(2 - \frac{1}{|\mathcal{Y}|})$, $C = \frac{2}{m}\left(1 - \frac{1}{|\mathcal{Y}|}\right)\left(4R_{\max}^2 + \frac{\lambda^2}{|\mathcal{X}|}\right) + d\sigma^2$, $G^2 = 1 - \frac{1}{|\mathcal{Y}|}$ and $F = 1$. $\qquad\square$

Note that the sensitivity $\Delta$ of the gradient estimator $\widetilde{\nabla}_m J_\lambda(\theta)$ is dominated by the data-dependent term. Despite introducing the regularization term $\lambda$, this term only depends on the policy parameters $\theta$ (independent of data), thus it does not affect the sensitivity. So the $\ell_2$-sensitivity of the gradient remains same as before.

**Lemma 9.** *let $\sigma^2 = \frac{16\ln(1.25/\delta)\cdot R_{\max}^2 G^2}{m^2\varepsilon^2}$, the batch size $m$ be set as: $m = (1/\varepsilon)^{2/3}(Nd)^{1/3}$, and $\eta = min(\frac{1}{LB}, \frac{\sqrt{2\delta_1}}{\sqrt{TLC}})$, we have:*

$$\mathbb{E}\left[\|\nabla J_\lambda(\theta_U)\|^2\right] \leqslant O\left(\frac{1}{\sqrt{N}} + \left(\frac{\sqrt{d}}{N\varepsilon}\right)^{2/3}\right). \tag{19}$$

*Proof.* for $\eta = min(\frac{1}{LB}, \frac{\sqrt{2\delta_1}}{\sqrt{TLC}})$ we know:

$$\mathbb{E}\left[\|\nabla J_\lambda(\theta_U)\|^2\right] \leqslant \frac{2\delta_1 LB}{T} + \frac{2\sqrt{2\delta_1 LC}}{\sqrt{T}} = O\left(\frac{1}{T} + \frac{\sqrt{C}}{\sqrt{T}}\right) = O\left(\frac{m}{N} + \frac{1}{\sqrt{N}} + \frac{\sigma\sqrt{md}}{\sqrt{N}}\right).$$

Plug in $\sigma^2 = \frac{16\ln(1.25/\delta)\cdot R_{\max}^2 G}{m^2\varepsilon^2}$ and $m = (1/\varepsilon)^{2/3}(Nd)^{1/3}$, we have:

$$\mathbb{E}\left[\|\nabla J_\lambda(\theta_U)\|^2\right] \leqslant O\left(\frac{1}{\sqrt{N}} + (\frac{\sqrt{d}}{N\varepsilon})^{2/3}\right).$$

$\qquad\square$

### E.3.2 Global optimum convergence

We first introduce an important proposition to bound our global private optimum convergence of softmax with log barrier regularization.

**Proposition 2** (Adapted from Theorem 5.2 in Agarwal et al. [5]). *Suppose $\theta$ is such that $\|\nabla J_\lambda(\theta)\| \leqslant \frac{\lambda}{2|\mathcal{X}||\mathcal{Y}|}$. Then for every initial distribution $\rho$, we have*

$$J^* - J(\theta) \leqslant 2\lambda. \tag{20}$$

*Proof.* Firstly, we define the following set of "bad" iterates:

$$I^+ \triangleq \left\{t \in \{1, \ldots, T\} \;\middle|\; \|\nabla J_\lambda(\theta_t)\| \geqslant \frac{\lambda}{2|\mathcal{X}||\mathcal{Y}|}\right\},$$

with $\lambda = \frac{\alpha}{2}$.

From Proposition 2, we know that if $\|\nabla J_\lambda(\theta)\| \leqslant \frac{\lambda}{2|\mathcal{X}||\mathcal{Y}|}$, we have $J^* - J(\theta) \leqslant 2\lambda$.

Hence, we have:

$$
\begin{aligned}
J^* - \frac{1}{T}\sum_{t=1}^{T} J(\theta_t) &= \frac{1}{T}\sum_{t\in I^+} J^* - J(\theta_t) + \frac{1}{T}\sum_{t\notin I^+} J^* - J(\theta_t) \\
&\overset{(a)}{\leqslant} \frac{|I^+|}{T}\cdot 4R_{\max} + \frac{1}{T}\sum_{t\notin I^+} J^* - J(\theta_t) \\
&\overset{(20)}{\leqslant} \frac{|I^+|}{T}\cdot 4R_{\max} + \frac{T-|I^+|}{T}\cdot 2\lambda \\
&\leqslant \frac{|I^+|}{T}\cdot 4R_{\max} + 2\lambda \\
&\leqslant \frac{|I^+|}{T}\cdot 4R_{\max} + \alpha,
\end{aligned}
\tag{21}
$$

where (a) holds by $J(\cdot) \leqslant 2R_{\max}$, then $J^* - J(\theta_t) \leqslant J^* + J(\theta_t) \leqslant 4R_{\max}$.

Now we turn to bound $|I^+|$. From the definition, we have:

$$
\sum_{t=1}^{T}\|\nabla J_\lambda(\theta_t)\|^2 \geqslant \sum_{t\in I^+}\|\nabla J_\lambda(\theta_t)\|^2 \geqslant \frac{|I^+|\lambda^2}{4|\mathcal{X}|^2|\mathcal{Y}|^2}.
$$

Through a straightforward mathematical transformation, we get

$$
\begin{aligned}
\frac{|I^+|}{T} &\leqslant \frac{4|\mathcal{X}|^2|\mathcal{Y}|^2}{\lambda^2}\cdot\frac{1}{T}\sum_{t=1}^{T}\|\nabla J_\lambda(\theta_t)\|^2 \\
&= \frac{16}{\alpha^2}\cdot|\mathcal{X}|^2|\mathcal{Y}|^2\cdot\frac{1}{T}\sum_{t=1}^{T}\|\nabla J_\lambda(\theta_t)\|^2.
\end{aligned}
$$

Thus, we have

$$
J^* - \frac{1}{T}\sum_{t=1}^{T} J(\theta_t) \overset{(21)}{\leqslant} \frac{64R_{\max}}{\alpha^2}|\mathcal{X}|^2|\mathcal{Y}|^2\cdot\frac{1}{T}\sum_{t=1}^{T}\|\nabla J_\lambda(\theta_t)\|^2 + \alpha.
$$

Taking expectation over the iterations on both sides, we have

$$
J^* - \frac{1}{T}\sum_{t=1}^{T}\mathbb{E}\left[J(\theta_t)\right] \leqslant \frac{64R_{\max}}{\alpha^2}|\mathcal{X}|^2|\mathcal{Y}|^2\cdot\frac{1}{T}\sum_{t=1}^{T}\mathbb{E}\left[\|\nabla J_\lambda(\theta_t)\|^2\right] + \alpha.
$$

To guarantee that $J^* - \frac{1}{T}\sum_{t=1}^{T}\mathbb{E}\left[J(\theta_t)\right] \leqslant \alpha$, we need to show:

$$
\frac{1}{T}\sum_{t=1}^{T}\mathbb{E}\left[\|\nabla J_\lambda(\theta_t)\|^2\right] \leqslant \alpha^3,
$$

Obviously, we have:

$$
E\left[\|\nabla J_\lambda(\theta_U)\|^2\right] = \frac{1}{T}\sum_{t=1}^{T}\mathbb{E}\left[\|\nabla J_\lambda(\theta_t)\|^2\right].
$$

Hence, based on Lemma 9, it is obvious to show that:

$$N \geqslant O\left(\frac{1}{\alpha^6} + \frac{\sqrt{d}}{\alpha^{9/2}\varepsilon}\right).$$

$\square$

# F    Proof of Chapter 6

## F.1    Proof of Theorem 4

For notation simplicity, we let $\pi_t = \pi_{\theta_t}$. By the performance difference lemma, we have

$$\sum_{t=1}^{T} J(\pi^*) - J(\pi_t) = \sum_{t=1}^{T} \mathbb{E}_{x \sim \rho, y \sim \pi^*(\cdot|x)} \left[A^{\pi_{\theta_t}}(x, y)\right].$$

Define $\mathrm{err}_t^* := \mathbb{E}_{x \sim \rho, y \sim \pi^*(\cdot|x)} \left[\left(A^{\pi_{\theta_t}}(x, y) - w_t^\top \nabla \log \pi_{\theta_t}(y \mid x)\right)\right]$. Then, we have

$$
\begin{aligned}
\sum_{t=1}^{T} J(\pi^*) - J(\pi_t) &= \sum_{t=1}^{T} \mathbb{E}_{x \sim \rho, y \sim \pi^*(\cdot|x)} \left[A^{\pi_t}(x, y)\right] \\
&= \sum_{t=1}^{T} \mathbb{E}_{x \sim \rho, y \sim \pi^*(\cdot|x)} \left[\langle w_t, \nabla_\theta \log \pi_t(y \mid x)\rangle\right] + \sum_{t=1}^{T} \mathrm{err}_t^* \\
&\stackrel{(a)}{=} \sum_{t=1}^{T} \mathbb{E}_{x \sim \rho, y \sim \pi^*(\cdot|x)} \left[\frac{1}{\eta}\langle \theta_{t+1} - \theta_t, \nabla_\theta \log \pi_t(y \mid x)\rangle\right] + \sum_{t=1}^{T} \mathrm{err}_t^* \\
&\stackrel{(b)}{\leqslant} \sum_{t=1}^{T} \mathbb{E}_{x \sim \rho, y \sim \pi^*(\cdot|x)} \left[\frac{1}{\eta}\log\left(\frac{\pi_{t+1}(y \mid x)}{\pi_t(y \mid x)}\right)\right] + \sum_{t=1}^{T} \frac{\eta\beta}{2}\|w_t\|^2 + \sum_{t=1}^{T} \mathrm{err}_t^* \\
&\stackrel{(c)}{\leqslant} \mathbb{E}_{x \sim \rho, y \sim \pi^*(\cdot|x)} \left[\frac{1}{\eta}\log\left(\frac{\pi_{T+1}(y \mid x)}{\pi_1(y \mid x)}\right)\right] + \frac{T\eta\beta}{2}W^2 + \sum_{t=1}^{T} \mathrm{err}_t^* \\
&\stackrel{(d)}{\leqslant} \frac{1}{\eta}\log|\mathcal{Y}| + \frac{T\eta\beta}{2}W^2 + \sum_{t=1}^{T} \mathrm{err}_t^*,
\end{aligned}
$$

where (a) holds by the update rule of our algorithm; (b) is true since the $\beta$-smooth condition of $\log \pi_\theta(y|x)$ is equivalent to the following inequality:

$$\forall \theta, \theta', x, y: \quad |\log \pi_{\theta'}(y \mid x) - \log \pi_\theta(y \mid x) - \nabla \log \pi_\theta(y \mid x) \cdot (\theta' - \theta)| \leqslant \frac{\beta}{2}\|\theta - \theta'\|_2^2;$$

(c) follows from the Assumption 5, which has a bounded norm of $W$, along with telescope sum; (d) is true since $\pi_1$ is a uniform distribution at each state. Thus, dividing by T on both sides and choosing $\eta = \sqrt{\frac{2\log|\mathcal{Y}|}{T\beta W^2}}$, yields

$$
\begin{aligned}
J(\pi^*) - \frac{1}{T}\sum_{t=1}^{T} J(\pi_t) &\leqslant \frac{\log|\mathcal{Y}|}{\eta T} + \frac{\eta\beta W^2}{2} + \frac{1}{T}\sum_{t=1}^{T} \mathrm{err}_t^* \\
&\leqslant \sqrt{\frac{\beta W^2 \log|\mathcal{Y}|}{2T}} + \frac{1}{T}\sum_{t=1}^{T} \mathrm{err}_t^*.
\end{aligned}
$$

To bound $\mathrm{err}_t^*$, we will simply leverage the guarantee of the regression oracle and the concentrability coefficient to transfer from $\mu$ to $\pi^*$. In particular, we have for any $t \in [T]$

$$
\begin{aligned}
\mathrm{err}_t^* &= \mathbb{E}_{x \sim \rho, y \sim \pi^*(\cdot|x)} \left[ \left( A^{\pi_{\theta_t}}(x, y) - w_t^\top \nabla \log \pi_{\theta_t}(y \mid x) \right) \right] \\
&\overset{(a)}{\leqslant} \sqrt{\mathbb{E}_{x \sim \rho, y \sim \pi^*(\cdot|x)} \left[ \left( A^{\pi_{\theta_t}}(x, y) - w_t^\top \nabla \log \pi_{\theta_t}(y \mid x) \right)^2 \right]} \\
&\overset{(b)}{\leqslant} \sqrt{C_{\mu \to \pi^*} \mathbb{E}_{x \sim \rho, y \sim \mu(\cdot|x)} \left[ \left( A^{\pi_{\theta_t}}(x, y) - w_t^\top \nabla \log \pi_{\theta_t}(y \mid x) \right)^2 \right]} \\
&\overset{(c)}{\leqslant} \sqrt{C_{\mu \to \pi^*} \cdot \mathrm{err}_t^2(m, \varepsilon, \delta, \zeta)},
\end{aligned}
$$

where (a) holds by Cauchy–Schwarz inequality; in (b), we define the single-policy concentrability coefficient $C_{\mu \to \pi^*} := \max_{x,y} \frac{\pi^*(y|x)}{\mu(y|x)}$; (c) follows directly from the guarantee of `PrivateLS` oracle.

Finally, putting everything together, yields

$$
J(\pi^*) - \frac{1}{T} \sum_{t=1}^{T} J(\pi_t) \leqslant \sqrt{\frac{\beta W^2 \log |\mathcal{Y}|}{2T}} + \frac{\sqrt{C_{\mu \to \pi^*}}}{T} \sum_{t=1}^{T} \mathrm{err}_t(m, \varepsilon, \delta, \zeta).
$$

### F.2 Proof of Lemma 2

A key lemma in our proof is the following form of Freedman's inequality.

**Lemma 10** (Lemma A.2 in [75]). *Let $\{X_i\}_{i \leq n}$ be a real-valued martingale difference sequence adapted to a filtration $\{\mathcal{F}_i\}_{i \leq n}$. If $|X_i| \leq R$ almost surely, then for any $\eta \in (0, 1/R)$, with probability at least $1 - \zeta$,*

$$
\sum_{i=1}^{n} X_i \leq \eta \sum_{i=1}^{n} \mathbb{E}_{i-1}[X_i^2] + \frac{\log(1/\zeta)}{\eta},
$$

*where $\mathbb{E}_{i-1}[\cdot] := \mathbb{E}[\cdot | \mathcal{F}_{i-1}]$.*

*Proof of Lemma 2.* For any fixed $h \in \mathcal{H}$, we define

$$
U_i^h := (h(u_i) - z_i)^2 - (h^*(u_i) - z_i)^2.
$$

If we define the filtration $\mathcal{F}_i = \sigma(u_1, z_1, \ldots, u_i, z_i)$ and let $\mathbb{E}_{i-1}[\cdot] = \mathbb{E}[\cdot | \mathcal{F}_{i-1}]$, then we have that $\{D_i^h\}_{i \leq m}$ where

$$
D_i^h := \mathbb{E}_{i-1}[U_i^h] - U_i^h
$$

is a martingale difference sequence adapted to $\{\mathcal{F}_i\}_{i \leq m}$. We further notice that

$$
\begin{aligned}
\mathbb{E}_{i-1}[(D_i^h)^2] &\leq \mathbb{E}_{i-1}[(U_i^h)^2] = \mathbb{E}_{i-1}[(h(u_i) - h^*(u_i))^2 (h(u_i) + h^*(u_i) - 2z_i)^2] \\
&\lesssim R^2 \cdot \mathbb{E}_{i-1}[(h(u_i) - h^*(u_i))^2],
\end{aligned}
$$

where the last step holds by the boundedness of $z_i$, $h \in \mathcal{H}$ and $h^*$. Moreover, by definition, we have

$$
\begin{aligned}
\mathbb{E}_{i-1}[U_i^h] &= \mathbb{E}_{i-1}[(h(u_i) - h^*(u_i))(h(u_i) + h^*(u_i) - 2z_i)] \\
&= \mathbb{E}_{i-1}[(h(u_i) - h^*(u_i))^2].
\end{aligned}
$$

With the above results, we first apply Lemma 10 to $\{D_i^h\}_{i \leq m}$ along with a union bound, yielding that with probability at least $1 - \zeta$, for all $h \in \mathcal{H}$

$$
\sum_{i=1}^{m} \mathbb{E}_{i-1}[(h(u_i) - h^*(u_i))^2] \lesssim \sum_{t=i}^{m} U_i^h + R^2 \cdot \log(|\mathcal{H}|/\zeta). \tag{22}
$$

Similarly, we can apply Lemma 10 to $\{-D_i^h\}_{i \leq m}$ along with a union bound, which give us

$$
\sum_{i=1}^{m} U_i^h \lesssim \sum_{i=1}^{m} \mathbb{E}_{i-1}[(h(u_i) - h^*(u_i))^2] + \log(|\mathcal{H}|/\zeta). \tag{23}
$$

Now, we set $h = \widehat{h}$ in (22), i.e., the output of the exponential mechanism, and by the standard utility guarantee of the exponential mechanism [34], we have

$$\sum_{i=1}^{m} \mathbb{E}_{i-1}[(\widehat{h}(u_i) - h^*(u_i))^2] \lesssim \sum_{i=1}^{m} U_t^{h'} + R^2 \log(|\mathcal{H}|/\zeta) + R^2 \frac{\log(|\mathcal{H}|/\zeta)}{\varepsilon},$$

where $h' \in \arg\min_{h \in \mathcal{H}} L(h) = \arg\min_{h \in \mathcal{H}} \sum_{i \in [m]} (h(u_i) - z_i)^2$. Since $\sum_{i=1}^{m} U_i^{h'} \le \sum_{i=1}^{m} U_i^{\widetilde{h}}$ where $\widetilde{h} := \operatorname{argmin}_{h \in \mathcal{H}} \sum_{i=1}^{m} \mathbb{E}_{i-1}[(h(u_i) - h^*(u_i))^2]$, by (23) and the above inequality, we have

$$\sum_{i=1}^{m} \mathbb{E}_{i-1}[(\widehat{h}(u_i) - h^*(u_i))^2] \lesssim \sum_{i=1}^{m} \mathbb{E}_{i-1}[(\widetilde{h}(u_i) - h^*(u_i))^2] + R^2 \log(|\mathcal{H}|/\zeta) + R^2 \frac{\log(|\mathcal{H}|/\zeta)}{\varepsilon}$$

$$\overset{(a)}{\le} m\alpha_{\mathsf{approx}} + R^2 \log(|\mathcal{H}|/\zeta) + R^2 \frac{\log(|\mathcal{H}|/\zeta)}{\varepsilon},$$

where $(a)$ holds the assumption on the approximation error. $\qquad\square$

## G    Proof of Appendix D

**Lemma 11.** *Consider any $t \in [T]$. For notation simplicity, we define $f_t(x, y) := \frac{1}{\eta} ln \frac{\pi_{t+1}(y|x)}{\pi_t(y|x)}$. Define $\Delta(x, y) = f_t(x, y) - r(x, y)$. Define $\Delta_{\pi_t}(x) = \mathbb{E}_{y \sim \pi_t(\cdot|x)} \Delta(x, y)$ and $\Delta_\mu(x) = \mathbb{E}_{y \sim \mu(\cdot|x)} \Delta(x, y)$. Under Assumption 6, for all $t$, we have the following:*

$$\mathbb{E}_{x, y \sim \pi_t(\cdot|x)} \left(f_t(x, y) - r(x, y) - \Delta_{\pi_t}(x)\right)^2 \le \mathrm{err}_t^2(m, \varepsilon, \delta, \zeta),$$
$$\mathbb{E}_{x, y \sim \mu(\cdot|x)} \left(f_t(x, y) - r(x, y) - \Delta_\mu(x)\right)^2 \le \mathrm{err}_t^2(m, \varepsilon, \delta, \zeta),$$
$$\mathbb{E}_x \left(\Delta_{\pi_t}(x) - \Delta_\mu(x)\right)^2 \le \mathrm{err}_t^2(m, \varepsilon, \delta, \zeta).$$

*Proof.* From Assumption 6, we have:

$$\mathbb{E}_{x, y_1 \sim \pi_t, y_2 \sim \mu} \left[ \left(f_t(x, y_1) - r(x, y_1) - \Delta_{\pi_t}(x)\right) - \left(f_t(x, y_2) - r(x, y_2) - \Delta_\mu(x)\right) + \Delta_{\pi_t}(x) - \Delta_\mu(x) \right]^2$$

$$= \mathbb{E}_{x, y_1 \sim \pi_t} \left(f_t(x, y_1) - r(x, y_1) - \Delta_{\pi_t}(x)\right)^2 + \mathbb{E}_{x, y_2 \sim \mu} \left(f_t(x, y_2) - r(x, y_2) - \Delta_\mu(x)\right)^2$$
$$- 2 \mathbb{E}_{x, y_1 \sim \pi_t, y_2 \sim \mu} \left(f_t(x, y_1) - r(x, y_1) - \Delta_{\pi_t}(x)\right) \left(f_t(x, y_2) - r(x, y_2) - \Delta_\mu(x)\right)$$
$$+ 2 \mathbb{E}_{x, y_1 \sim \pi_t} \left(f_t(x, y_1) - r(x, y_1) - \Delta_{\pi_t}(x)\right) \left(\Delta_{\pi_t}(x) - \Delta_\mu(x)\right)$$
$$- 2 \mathbb{E}_{x, y_2 \sim \mu} \left(f_t(x, y_2) - r(x, y_2) - \Delta_\mu(x)\right) \left(\Delta_{\pi_t}(x) - \Delta_\mu(x)\right)$$
$$+ \mathbb{E}_x \left(\Delta_{\pi_t}(x) - \Delta_\mu(x)\right)^2$$

$$= \mathbb{E}_{x, y_1 \sim \pi_t} \left(f_t(x, y_1) - r(x, y_1) - \Delta_{\pi_t}(x)\right)^2 + \mathbb{E}_{x, y_2 \sim \mu} \left(f_t(x, y_2) - r(x, y_2) - \Delta_\mu(x)\right)^2$$
$$+ \mathbb{E}_x \left(\Delta_{\pi_t}(x) - \Delta_\mu(x)\right)^2$$

$$\le \mathrm{err}_t^2(m, \varepsilon, \delta, \zeta).$$

In that case, since the total sum is less than $\mathrm{err}_t^2(m, \varepsilon, \delta, \zeta)$, it follows that each term must be less than $\mathrm{err}_t^2(m, \varepsilon, \delta, \zeta)$. Hence, the lemma holds. $\qquad\square$

**Lemma 12.** *Assume $\max_{x, y, t} |A_t(x, y)| \le A \in \mathbb{R}^+$, and $\pi_1$ is uniform over $\mathcal{Y}$. Then with $\eta = \sqrt{\ln(|\mathcal{Y}|)/(A^2 T)}$, for the sequence of policies computed by REBEL, we have:*

$$\forall \pi, x : \sum_{t=1}^{T} \mathbb{E}_{y \sim \pi(\cdot|x)} A_t(x, y) \le 2A\sqrt{\ln(|\mathcal{Y}|)T}.$$

*Proof.* By the definition of $f_t$, we have

$$\Delta(x, y) = \frac{1}{\eta} \ln \frac{\pi_{t+1}(y|x)}{\pi_t(y|x)} - r(x, y).$$

Taking $\exp$ on both sides, we get:

$$\forall x, y: \quad \pi_{t+1}(y|x) = \pi_t(y|x) \exp\left(\eta\left(r(x, y) + \Delta(x, y)\right)\right) = \frac{\pi_t(y|x) \exp\left(\eta(r(x, y) + \Delta(x, y) - \Delta_\mu(x))\right)}{\exp(-\eta\Delta_\mu(x))}.$$

Denote

$$g_t(x, y) := r(x, y) + \Delta(x, y) - \Delta_\mu(x),$$

and the advantage

$$A_t(x, y) = g_t(x, y) - \mathbb{E}_{y' \sim \pi_t(\cdot|x)} g_t(x, y').$$

We can rewrite the above update rule as:

$$\forall x, y: \quad \pi_{t+1}(y|x) \propto \pi_t(y|x) \exp\left(\eta A_t(x, y)\right)$$

The remain part of the proof is similar to the analysis of NPG in F.1. $\qquad\square$

### G.1 Proof of Theorem 6

*Proof.* We know that:

$$\frac{1}{T} \sum_{t=1}^{T} \left(\mathbb{E}_{x, y \sim \pi^*(\cdot|x)} r(x, y) - \mathbb{E}_{x, y \sim \pi_t(\cdot|x)} r(x, y)\right) = \frac{1}{T} \sum_{t=1}^{T} \mathbb{E}_{x, y \sim \pi^*(\cdot|x)} \left(A^{\pi_t}(x, y)\right).$$

Then, we have:

$$
\begin{aligned}
\frac{1}{T} \sum_{t=1}^{T} \mathbb{E}_{x, y \sim \pi^*(\cdot|x)} \left(A^{\pi_t}(x, y)\right) &= \frac{1}{T} \sum_{t=1}^{T} \mathbb{E}_{x, y \sim \pi^*(\cdot|x)} \left(A_t(x, y)\right) \\
&\quad + \frac{1}{T} \sum_{t=1}^{T} \mathbb{E}_{x, y \sim \pi^*(\cdot|x)} \left(A^{\pi_t}(x, y) - A_t(x, y)\right) \\
&\stackrel{(a)}{\leqslant} 2A\sqrt{\frac{\ln(|\mathcal{Y}|)}{T}} \\
&\quad + \frac{1}{T} \sum_{t=1}^{T} \sqrt{\mathbb{E}_x \mathbb{E}_{y \sim \pi^*(\cdot|x)} \left(A^{\pi_t}(x, y) - A_t(x, y)\right)^2},
\end{aligned}
$$

where (a) holds by Lemma 12.

Then we need to bound: $\mathbb{E}_x \mathbb{E}_{y \sim \pi^*(\cdot|x)} \left(A^{\pi_t}(x, y) - A_t(x, y)\right)^2$.

By the definition of concentrability coefficient $C_{\mu \to \pi^*}$, we know that:

$$\mathbb{E}_x \mathbb{E}_{y \sim \pi^*(\cdot|x)} (A^{\pi_t}(x, y) - A_t(x, y))^2 \leqslant C_{\mu \to \pi^*} \mathbb{E}_{x, y \sim \mu(\cdot|x)} (A^{\pi_t}(x, y) - A_t(x, y))^2$$

We now bound $\mathbb{E}_{x, y \sim \mu(\cdot|x)} (A^{\pi_t}(x, y) - A_t(x, y))^2$.

$$
\begin{aligned}
&\mathbb{E}_{x, y \sim \mu(\cdot|x)} (A^{\pi_t}(x, y) - A_t(x, y))^2 \\
&= \mathbb{E}_{x, y \sim \mu(\cdot|x)} (r(x, y) - \mathbb{E}_{y' \sim \pi_t(\cdot|x)} r(x, y') - g_t(x, y) + \mathbb{E}_{y' \sim \pi_t(\cdot|x)} g_t(x, y'))^2 \\
&\leqslant 2\mathbb{E}_{x, y \sim \mu(\cdot|x)} \left(r(x, y) - g_t(x, y)\right)^2 + 2\mathbb{E}_x \mathbb{E}_{y' \sim \pi_t(\cdot|x)} \left(r(x, y') - g_t(x, y')\right)^2
\end{aligned}
$$

Recall the $g_t(x, y) = r(x, y) + \Delta(x, y) - \Delta_\mu(x)$, and from Lemma 11, we can see that

$$\mathbb{E}_{x,y \sim \mu(\cdot|x)}(r(x, y) - g_t(x, y))^2 = \mathbb{E}_{x,y \sim \mu(\cdot|x)}(\Delta(x, y) - \Delta_\mu(x))^2 \leqslant \text{err}_t^2(m, \varepsilon, \delta, \zeta).$$

For $\mathbb{E}_x \mathbb{E}_{y' \sim \pi_t(\cdot|x)}(r(x, y') - g_t(x, y'))^2$, we have:

$$\begin{aligned}
\mathbb{E}_x \mathbb{E}_{y' \sim \pi_t(\cdot|x)}(r(x, y') - g_t(x, y'))^2 &= \mathbb{E}_x \mathbb{E}_{y' \sim \pi_t(\cdot|x)}(\Delta(x, y') - \Delta_\mu(x))^2 \\
&= \mathbb{E}_x \mathbb{E}_{y' \sim \pi_t(\cdot|x)}(\Delta(x, y') - \Delta_{\pi_t}(x) + \Delta_{\pi_t}(x) - \Delta_\mu(x))^2 \\
&\leqslant 2\mathbb{E}_x \mathbb{E}_{y' \sim \pi_t(\cdot|x)}(\Delta(x, y') - \Delta_{\pi_t}(x))^2 + 2\mathbb{E}_x(\Delta_{\pi_t}(x) - \Delta_\mu(x))^2 \\
&\leqslant 4\text{err}_t^2(m, \varepsilon, \delta, \zeta),
\end{aligned}$$

where the last inequality uses Lemma 11 again.

Combining things together, we can conclude that:

$$\mathbb{E}_x \mathbb{E}_{y \sim \pi^*(\cdot|x)}(A^{\pi_t}(x, y) - A_t(x, y))^2 \leqslant C_{\mu \to \pi^*}(10\text{err}_t^2(m, \varepsilon, \delta, \zeta)).$$

Hence, we can derive our main theorem:

$$\begin{aligned}
\frac{1}{T}\sum_{t=1}^{T} \mathbb{E}_{x,y \sim \pi^*(\cdot|x)}(A^{\pi_t}(x, y)) &\leqslant 2A\sqrt{\frac{\ln|\mathcal{Y}|}{T}} + \frac{1}{T}\sum_t \sqrt{10C_{\mu \to \pi^*}\text{err}_t^2(m, \varepsilon, \delta, \zeta)} \\
&= 2A\sqrt{\frac{\ln|\mathcal{Y}|}{T}} + \frac{\sqrt{10C_{\mu \to \pi^*}}}{T}\sum_{t=1}^{T}\text{err}_t(m, \varepsilon, \delta, \zeta).
\end{aligned}$$

$\square$

## H   Experiments

**Environment.** We conduct experiments on the `CartPole-v1` environment from OpenAI Gym, a standard benchmark for evaluating policy gradient methods. The task requires balancing a pole on a moving cart, with a maximum episode reward of 500.

**Policy Parameterization.** Policies are represented by two-layer fully-connected neural networks with 64 hidden units, ReLU activation, and softmax output layer, i.e., the architecture is $\text{Linear}(4, 64) \to \text{ReLU} \to \text{Linear}(64, 2) \to \text{Softmax}$.

**Privacy Settings.** We evaluate privacy-preserving algorithms under two privacy budgets: $(\varepsilon, \delta) = (5, 10^{-5})$ and $(\varepsilon, \delta) = (3, 10^{-5})$, representing moderate and strong privacy guarantees respectively.

**Training Details.** All algorithms are trained for 100 epochs with batch size 10 (i.e., 10 episodes per gradient update) and discount factor $\gamma = 0.99$. We use advantage normalization with baseline subtraction for variance reduction. Each algorithm is trained with 3 random seeds, and we report the average performance with standard deviation.

**Evaluation.** We evaluate performance using three metrics: (i) **Mean Final Reward**: average reward in the final epoch across all seeds, (ii) **Std Final Reward**: standard deviation of final rewards across seeds, and (iii) **Best Epoch Mean**: highest average reward achieved during training.

**Results.** Table 1 summarizes the performance under different privacy settings.

We can see that (i) NPG consistently outperforms PG in both private and non-private settings, demonstrating the benefit of curvature-aware updates, (ii) `DP-NPG` with $\varepsilon = 5$ achieves near-optimal performance ($\sim 500$), aligning with our theoretical predictions and empirical findings in Rio et al. [13] who use PPO instead of NPG, and (iii) as privacy budget decreases (smaller $\varepsilon$), performance degrades as expected from theory.

| Algorithm | $\varepsilon$ | Mean Final Reward | Std Final Reward | Best Epoch Mean |
|-----------|---------------|-------------------|------------------|-----------------|
| PG | N/A | 334.37 | 25.25 | 361.70 |
| DP-PG | 5 | 190.34 | 52.91 | 199.17 |
| DP-PG | 3 | 143.87 | 22.88 | 187.17 |
| NPG | N/A | 492.90 | 10.04 | 500.00 |
| DP-NPG | 5 | 478.73 | 28.05 | 494.70 |
| DP-NPG | 3 | 400.87 | 65.37 | 410.43 |

Table 1: Performance summary for `CartPole-v1` under different privacy budgets ($\varepsilon = 5$ and $\varepsilon = 3$, $\delta = 10^{-5}$).

# I  Limitations

In this study, we propose private variants of three classical algorithms for policy optimization and provide a comprehensive analysis of their sampling complexity under both private and non-private settings. Our analysis successfully recovers the classical complexity bounds in the non-private regime, validating the theoretical soundness of our approach. However, our current results focus only on the one-pass sampling setting; the sampling complexity in the multi-pass scenario may admit further improvements.

Moreover, while this work primarily focuses on the theoretical foundations of the proposed algorithms, we have also conducted simple empirical validations to support our theoretical findings. It should be noted, however, that we have not yet performed large-scale evaluations on real-world datasets. Nevertheless, since our methods serve as core components in policy optimization, they have broad applicability across various reinforcement learning domains—particularly in privacy-sensitive settings such as reinforcement learning with human feedback (RLHF) and medical data analysis. Applying our approach to these areas could further enhance the secure handling of sensitive information.

