# OpenReview forum: "On the Sample Complexity of Differentially Private Policy Optimization"
_NeurIPS.cc/2025/Conference — NeurIPS 2025 poster_

### Official Review · Reviewer_LxTk · 2025-06-30

**Clarity:** 3
**Significance:** 3
**Originality:** 3
**Rating:** 4
**Confidence:** 2

**Summary:**

This paper initiates a theoretical study of differentially private policy optimization (PO), focusing on sample complexity under privacy constraints. The authors first define a tailored notion of differential privacy (DP) for PO, addressing challenges arising from on-policy learning dynamics and the definition of a privacy unit. The authors propose a meta-algorithm for private PO, instantiating it for policy gradient (DP-PG), natural policy gradient (DP-NPG), and REBEL (DP-REBEL). Theoretical results establish polynomial sample complexity bounds, showing that privacy costs often appear as lower-order terms, and the analysis is validated through corollaries for log-linear policies in low and high dimensions.

**Questions:**

1. Could the authors detail the technical difficulties in deriving sample complexity bounds?
2. Will empirical results on RL benchmarks be provided to validate the practical performance of the proposed methods?
3. Could the authors give any insight on how their framework handles multi-round user interactions, which may violate the single-sample privacy unit assumption?

**Ethical Concerns:**

["NO or VERY MINOR ethics concerns only"]

**Final Justification:**

My concerns regarding the experiment and technical contribution have been addressed, and I will keep my score at 4.

**Limitations:**

Yes.

**Paper Formatting Concerns:**

There is no major formatting issue.

**Quality:**

3

**Strengths And Weaknesses:**

## Strengths
1. This paper is well-organized and easy to follow.
2. The study connects theory to real-world applications (e.g., healthcare, LLM alignment), where privacy is critical. The examples of LLM finetuning and healthcare-rl help me better understand the broad application of differential privacy policy optimization.
3. The authors derive rigorous sample complexity bounds under various settings. Key contributions include: 1) Propose a general framework for privacy policy optimization. 2) Proving that privacy costs manifest as lower-order terms in sample complexity, e.g., $\(O(1/\alpha^4)\)$ vs. $\(O(\sqrt{d}/(\alpha^3\varepsilon))\)$. 3) Showing how structural properties (e.g., Fisher non-degeneracy, compatibility) refine regret guarantees.

## Weaknesses
1. The proposed framework and theoretical analysis are not highly innovative, which mainly combines existing policy optimization algorithms (e.g., PG, NPG) with differential privacy techniques.
2. The paper lacks experimental results on real-world benchmarks.

---

> ### Author Rebuttal · Authors · 2025-07-30
>
> Thanks for your positive evaluation of our paper. We will recap your valuable comments and present our detailed response.
>
> **1. Technical difficulty.** One technical contribution worth highlighting is Lemma 2, which gives the first estimation error bound for private LS with the exponential mechanism under approximation error. This enables us to be the first to analyze the sample complexity of DP-NGP and DP-REBEL under **pure DP, even for general function approximations**. Moreover, given the wide use of private LS oracle in other RL settings, we believe that Lemma 2 is of broad interest to the community.
>
> **2. Experiments.** Yes, we have added some experiments on standard RL benchmarks to validate the practical performance of proposed algorithms, see our response to Reviewer XUJ9 above.
>
> **3. Multi-round user interactions.** This is an insightful and sharp point. Our current framework assumes **one-round interaction per user**, which is standard in almost all prior work on DP-RL. To support **multi-round user interactions**, we distinguish two cases in the context of PO:
>
> - **(a) Within-batch reuse**: If a user appears multiple times within the same batch, then the sensitivity must scale with the number of appearances. We can incorporate this into the privacy analysis by adjusting the added noise.
>
> - **(b) Across-batch reuse**: If a user interacts in multiple rounds across different batches, we can apply **privacy composition theorems** (e.g., advanced composition or Rényi DP accounting) to compute cumulative privacy loss.

---

### Official Review · Reviewer_FwLs · 2025-07-02

**Clarity:** 3
**Significance:** 3
**Originality:** 3
**Rating:** 4
**Confidence:** 2

**Summary:**

While policy optimization is an important technique in RL and LLM, its privacy concerns could be an issue. This paper focuses on the sample complexity of private PO algorithms. They first analyze the challenges of defining DP-PO compared to standard SFT, and then formulate the definition of DP for PO. The sample complexity of three commonly used algorithms (PG, NPG, and REBEL) is studied under DP setting.

**Questions:**

- Suppose there is a dataset of size $N$. Does the user represent a static prompt? If so, then why is the user dynamic?

- Do current reward models satisfy the assumption of bounded reward $[R_{min}, R_{max}]$?

- In Theorem 4, $|Y|$ is typically the same as the vocabulary size in LLMs, which is very large. Will the bound be loose in that case?

  Minor:

  - Eq.1: it should be $y \sim \pi_{\theta}(\cdot|x)$

**Ethical Concerns:**

["NO or VERY MINOR ethics concerns only"]

**Final Justification:**

The authors have addressed most of my concerns and I will maintain my support to the paper.

**Limitations:**

Yes

**Quality:**

3

**Strengths And Weaknesses:**

Strength:

- The motivation to analyze DP-PO with sample complexity is well discussed. The paper also gives a detailed discussion on related work, leading to a clear demonstration of its contribution and novelty.
- While DP can protect privacy, it often requires extra cost. As PO needs to sample multiple trajectories, the sample complexity is especially critical. This paper targets this interesting and important question, which would benefit the community of both PO and DP.
- This paper proposes a unified framework to analyze three popular and important algorithms, including PG, NPG, and REBEL. The sample complexity is also studied in various conditions, such as FOSP and global optimum.

Weakness:

- The formulation of PO in Sec. 2 maximizes the reward. However, in practice, a constraint on the KL divergence of $\pi_{\theta}$ and $\pi_{SFT}$ is usually used. Ignoring this term would limit the applicability of this paper.
- The definition of “DP in PO” is not clear enough to me. In Def.2 the author uses the term “users” to define DP-PO, while the description of users is overwhelming. I suggest giving more illustrations on the description of the users and adding more mathematical details in Def . 2.
- I wonder if the theories in this paper can be easily evaluated by (simulated) experiments. Empirical experiments will enhance the validity and application of the theories.

---

> ### Author Rebuttal · Authors · 2025-07-30
>
> Thanks for your positive evaluation of our paper. We will recap your valuable comments and present our detailed response.
>
> **1. Metric.** We clarify that our work adopts the **standard reward-maximization formulation** used in most prior work on PO, such as PG, NPG, REBEL, TRPO, and PPO. For LLM, we agree that we often add an additional KL term. In this case, our algorithm can still be implemented by merging the KL term into the reward.
>
> **2. DP in PO.** We can use Examples 1 and 2 in the paper to understand the concept of "users". Each "user" can represent each prompt in Example 1, while "user" can also represent each patient in Example 2. We clarify that the word "dynamic" refers to the dataset. That is,  as the reviewer mentioned, each user can be each prompt (which is the $x$ and it is static), but the response $y$ and reward $r(x,y)$ are dynamic, since they are determined in the on-policy manner. This is in sharp contrast to supervised learning where the whole dataset $(x_i,y_i)$ is static and fixed in advance.
>
>
> **3. Experiments.** Although our paper focuses on theory, we have now added empirical evaluations using the CartPole-v1 benchmark in OpenAI Gym. As detailed in our response to Reviewer XUJ9, the results validate the theoretical trends.
>
>
> **4. Parameters.** Yes, many reward models in practice satisfy boundedness, like in LLM alignment and reasoning. We also note that our dependence on $ |Y| $ is only on the **log order**, which is the same as the non-private case, like REBEL. Moreover, as shown in the REBEL paper, the practical performance is quite good even with a theory bound of $\log |Y|$.

---

> ### Author Response · Authors · 2025-08-07
> **A kind follow-up**
>
> Thank you again for your valuable comments!
>
> As the discussion period is drawing to a close, we wanted to check in and see if our rebuttal has helped clarify your concerns. We’d be happy to further engage if there are any remaining questions.
>
> If our response is satisfying, we would also appreciate knowing whether there’s an opportunity to update your score before the final justification.
>
> Best,
>
> Authors

---

### Official Review · Reviewer_VYZ7 · 2025-07-03

**Clarity:** 3
**Significance:** 3
**Originality:** 3
**Rating:** 5
**Confidence:** 2

**Summary:**

The paper studies DP version of 3 big algorithms in policy optimization in RL by stating suitable new DP definitions, proposing DP algorithms, and proving that adding the privacy requirement do not increase the sample complexity.

**Questions:**

- while the result looks good, should I think of the extra terms from privacy as "expected"? I.e. it is intuitive to say that with "epsilon noise", the penalty is around the same level as alpha? Maybe some toy examples or theoretical analysis of lower bound / tightness could help that this privacy term is expected or tight or not.

**Ethical Concerns:**

["NO or VERY MINOR ethics concerns only"]

**Final Justification:**

I see the authors adding some experimental results. I am not confident to judge them, but I do not think this is their main contribution anyway. My bigger concern though is the confidence among pool of reviewers we have.

**Quality:**

4

**Strengths And Weaknesses:**

(+) seems like a new, useful area of study. The main contribution must be the ideation of the settings (the alg or adding Gaussian is more standard and expected)
(+) complete statement and proofs
(-) as it is on theory, it's with no experiments and estimate of constants are out of scope here. It may lead to more practical studies later.

---

> ### Author Rebuttal · Authors · 2025-07-30
>
> Thank you for your positive evaluation and encouraging remarks. We are glad to hear that you found the problem setting and theoretical contributions promising for future work.
>
> **1. Privacy cost.** A key insight of our paper is that the additional sample complexity cost induced by privacy manifests as **lower-order terms** in many problem settings. That is, the $\alpha$ term related to privacy is smaller than the non-private $\alpha$ term, illustrating that privacy cost does not dominate the sample complexity.
>
> To complement the theory, we have now conducted experiments (see our response to Reviewer XUJ9) and observed that DP-NPG can nearly match non-private NPG performance even under privacy ( $\epsilon = 5$ ). These empirical results further support our theory.
>
> On the other hand, we also tend to believe that there exists a gap between our privacy term and its corresponding lower bound. We leave it as future work, as even for the supervised learning setting, there exist gaps between upper and lower bounds in the case of private non-convex optimization.

---

> > ### Comment · Reviewer_VYZ7 · 2025-08-04
> >
> > Thank you for your explanations.

---

### Official Review · Reviewer_XUJ9 · 2025-07-07

**Clarity:** 3
**Significance:** 2
**Originality:** 2
**Rating:** 3
**Confidence:** 3

**Summary:**

This paper is purely theoretical (it includes no experiments or algorithms) and derives sample complexity bounds for Differentially Private Policy Optimization (DP-PO). After redefining adjacent datasets in terms of users, the authors present a general scheme for DP-PO. First, the authors introduce a vanilla DP-PG algorithm and provide corresponding privacy guarantees and convergence bounds. Finally, they derive the DP-NPG and DP-REBEL algorithms, also with privacy guarantees and regret bounds.

**Questions:**

see weaknesses

**Ethical Concerns:**

["NO or VERY MINOR ethics concerns only"]

**Limitations:**

yes, in the appenx

**Quality:**

3

**Strengths And Weaknesses:**

Strengths
- The paper is well written and easy to follow
- The problem addressed in this paper is important for the DP-RL community. Finite-sample guarantees are particularly valuable for understanding the impact of various parameters and could help guide parameter tuning.
- The theoretical analysis is exhaustive thoroughly covers privacy guarantees, convergence, and regret bounds across different algorithmic settings

Weaknesses

- My main concern about this paper is the lack of experimental results. A key challenge in DP algorithms is selecting the appropriate \epsilon which requires balancing performance and privacy.For the three proposed algorithms, it is difficult to gain intuition about reasonable values for this parameter. Have the authors conducted any experiments, even on small benchmarks like RiverSwim, to develop intuition? What happens when using large values for the parameter \w, for example in the case of a deep policy?
- DP-PG seems to rely on a parameter G that appears to be impossible to compute directly. Can the authors propose a practical method for setting this parameter? What are the theoretical implications of adopting a clipping strategy similar to that used in DP-SGD? Once again, an empirical evaluation of this parameter’s impact would have been highly beneficial.
- Although the theoretical work is novel, I don't find enough originality in the paper: defining DP in an RL context has already been analyzed and the proposed algorithms do not differ a lot from the work of [Rio et al, 2025].
- [Minor remark] Placing the section on DP-REBEL in the appendix disrupts the flow of the paper, alhtough the space constraint and trade-off.

---

> ### Author Rebuttal · Authors · 2025-07-30
>
> Thank you for your thoughtful and detailed review. Below, we address your key concerns point by point. We hope our answers will resolve your concern.
>
> **1. Experiments.**
> While the primary contribution of our paper is theoretical—providing the first sample complexity guarantees for DP policy optimization—we appreciate the reviewer's suggestion for empirical validation. We conducted experiments using the standard CartPole environment from OpenAI Gym with two-layer neural networks as policies.
> For DP-NPG, we implemented a simple differentially private least-squares update by adding calibrated Gaussian noise to the Fisher information matrix and bias vector.
>
>
> Since we are unable to upload images, we summarize the results using the table below:
>
> ### Performance Summary for CartPole-v1( $\epsilon = 5$  and $\epsilon = 3$ with $\delta = 1e-5$)
>
> | Algorithm | $\epsilon$ | Mean Final Reward | Std Final Reward | Best Epoch Mean |
> |-----------|---|-------------------|------------------|-----------------|
> | PG        | NA | 334.37            | 25.25            | 361.70          |
> | DP-PG     | 5 | 190.34            | 52.91            | 199.17          |
> | DP-PG     | 3 | 143.87            | 22.88            | 187.17          |
> | NPG       | NA | 492.90            | 10.04            | 500.00          |
> | DP-NPG    | 5 | 478.73            | 28.05            | 494.70          |
> | DP-NPG    | 3 | 400.87            | 65.37            | 410.43          |
>
>
> **Takeaway messages:**
> - (a) NPG consistently outperforms PG, both in private and non-private settings, thanks to its curvature-aware updates.
>
> - (b) DP-NPG achieves near-optimal reward (~500) even with privacy ($\epsilon$ = 5), consistent with our theory (i.e., privacy cost could be a lower-order term) as well as the empirical results in [Rio et al., 2025], which uses PPO instead of NPG. Note that, given the close relationship between NPG (TRPO) and PPO, this is expected.
>
> - (c) As $\epsilon$ decreases (i.e., stronger privacy), utility degrades, as expected from our theory.
>
> ---
>
> **2. Clipping norm.** This parameter is regarded as a hyperparameter in DP ML, even in DP-SGD for supervised leanring settings.
> There are two common approaches for choosing it (see [R1]):
> - Use a small amount of public data to estimate the norm
> - Jointly tune it with the learning rate
>
> We combine the two approaches when determining our clipping norm. We finally mention in passing that another possible approach could be leveraging the insight derived in [Rio et al., 2025] when choosing this parameter.
>
> [R1] Ponomareva, Natalia, et al. "How to dp-fy ml: A practical guide to machine learning with differential privacy." Journal of Artificial Intelligence Research 77 (2023): 1113-1201.
>
> ---
>
> **3. Comparison with [Rio et al, 2025].**
> While [Rio et al., 2025] makes an important empirical contribution, our work is the **first to establish formal sample complexity bounds** for DP versions of PG, NPG, and REBEL. Our unified framework complements prior empirical work and provides a theoretical foundation for principled privacy–utility tradeoffs in policy optimization.
>
>
>
> ---
>
> **4. Presentation.** We appreciate the suggestion and agree that DP-REBEL is a core contribution. We will move more discussion into the main text in the final version.

---

> > ### Author Response · Authors · 2025-08-07
> > **A kind follow-up**
> >
> > Thank you again for your valuable comments!
> >
> > As the discussion period is drawing to a close, we wanted to check in and see if our newly added experiment has helped clarify your concerns. We’d be happy to further engage if there are any remaining questions.
> >
> > If our response is satisfying, we would also appreciate knowing whether there’s an opportunity to update your score before the final justification.
> >
> > Best,
> >
> > Authors

---

### Comment · Area_Chair_tCiE · 2025-08-08

Dear Reviewers,

Thank you for reviewing the manuscript.

Please notice that having a comprehensive exchange with the authors is an obligation as the reviewers. It is not okay to simply click the acknowledgement button without any discussion. Please use the remaining two days to discuss with the authors and respond to the author rebuttal.

AC

---

### Decision · Program_Chairs · 2025-09-17

**Decision:**

Accept (poster)

**Comment:**

This work analyzes the sample complexity of several PO algorithms under DP constraints. There is one unified framework for doing this. The manuscript shows that the privacy costs can often manifest as lower-order terms in the sample complexity.

The reviews are mostly positive, with one borderline rejection review. While I inspect the reviews, I found the concerns have been mainly addressed in the rebuttal. I suggest the authors to incorporate the additional results they have provided in the rebuttal to their next revision. I recommend acceptance.